# Assimilation of SMOS brightness temperature into a large-scale distributed conceptual hydrological model to improve soil moisture predictions: the Murray-Darling basin in Australia as a test case.

Renaud Hostache[1], Dominik Rains[2,3], Kaniska Mallick[1], Marco Chini[1], Ramona Pelich[1], Hans Lievens[2,4], Fabrizio Fenicia[5], Giovanni Corato[1], Niko E.C. Verhoest[2], and Patrick Matgen[1]

[1]Luxembourg Institute of Science and Technology (LIST), Department Environmental Research and Innovation, Belvaux, Luxembourg
[2]Ghent University, Department of environment, Ghent, Belgium
[3]University of Leicester, Earth Observation Science, Department of Physics & Astronomy, Leicester, UK
[4]KU Leuven, Department of Earth and Environmental Sciences, Heverlee, Belgium
[5]Swiss Federal Institute of Aquatic Science and Technology (EAWAG), Department Systems Analysis, Integrated Assessment and Modelling, Dübendorf, Switzerland

**Correspondence:** Renaud Hostache (renaud.hostache@list.lu)

**Abstract.** The main objective of this study is to investigate how brightness temperature observations from satellite microwave sensors may help in reducing errors and uncertainties in soil moisture and evapotranspiration simulations with a large-scale conceptual hydro-meteorological model. In addition, this study aims to investigate whether such a conceptual modelling framework, relying on parameter calibration, can reach the performance level of more complex physically-based models for soil moisture simulations at a large scale. We use the ERA-Interim publicly available forcing dataset and couple the CMEM radiative transfer model with a hydro-meteorological model enabling therefore soil moisture, evapotranspiration and brightness temperature simulations over the Murray-Darling Basin in Australia. The hydro-meteorological model is configured using recent developments of the SUPERFLEX framework, which enables tailoring the model structure to the specific needs of the application as well as to data availability and computational requirements. The hydrological model is first calibrated using only a sample of SMOS brightness temperature observations (period 2010-2011). Next, SMOS brightness temperature observations are sequentially assimilated into the coupled SUPERFLEX-CMEM model (period 2010-2015). For this experiment, a Local Ensemble Transform Kalman Filter is used. Our empirical results show that the SUPERFLEX-CMEM modelling chain is capable of predicting soil moisture at a performance level similar to that obtained for the same study area and with a quasi-identical experimental set up using the CLM land surface model. This shows that a simple model, when calibrated using globally and freely available Earth observation data, can yield performance levels similar to those of a physically-based (uncalibrated) model. The correlation between simulated and *in situ* observed soil moisture ranges from 0.62 to 0.72 for the surface and root zone soil moisture. The assimilation of SMOS brightness temperature observations into the SUPERFLEX-CMEM modelling chain improves the correlation between predicted and *in situ* observed surface and root zone soil moisture by 0.03 on average, showing improvements similar to those obtained using the CLM land surface model. Moreover, the assimilation improves at the same time the correlation between predicted and *in situ* observed monthly evapotranspiration by 0.02 on average.

## 1 Introduction

Motivated by the impact of climate change on the scarcity or excess of water in many areas around the world, and following the recommendations of the Sendaï framework for disaster risk reduction (UNISDR, 2015), several agencies and research
institutions have put substantial efforts in better monitoring and predicting the hydrologic cycle at a global scale. Such monitoring/prediction efforts are indeed necessary for assessing the risk of extreme hydrological events and for enabling early warning (Revilla-Romero et al., 2016), especially considering that impacts related to such hydrological extremes are expected to increase in the future due to the combined effect of socio-economic development and climate change (Lehner et al., 2016).

Numerical models such as hydrological and land surface models are central to predict and forecast droughts (Rains et al.,
2017) and help in better anticipating disaster and the associated emergency response (Revilla-Romero et al., 2016). However, model simulations suffer from inherent uncertainties (Liu and Gupta, 2007) due to the simplified representation of physical processes as well as uncertain forcing (García-Pintado et al., 2015; Hostache et al., 2011) and the lack of data for setting up and controlling them (Pappenberger et al., 2007; Hostache et al., 2015; Wood et al., 2016). To reduce uncertainty in model simulations, an advanced solution that has gained increased interest over the last decades is the integration of remote sensing data
into models (Andreadis and Schumann, 2014; Hostache et al., 2018; De Lannoy and Reichle, 2016b). This approach pursues an optimal combination of hydro-meteorological modelling and remote sensing, for example by using satellite measurements as forcing or calibration data and/or for regularly updating the model states or parameters (Moradkhani, 2007). This allows periodically controlling and correcting the models via external observations. In forecasting mode, such data assimilation approaches allow keeping the predictions on track, while in hind-casting mode they enable improved simulations of measured
fluxes and states of the past.

Many advances have been made in these areas of research and spaceborne sensors are already providing a wealth of Earth observation data with many applications in hydrology (Brocca et al., 2012; De Lannoy and Reichle, 2016b). In particular, satellite surface soil moisture (SM) estimates are available at temporal and spatial resolutions compatible with operational hydrology requirements especially at the large scale (De Lannoy and Reichle, 2016b). Although the assimilation of *in situ* data
is widely established in operational hydrology (Ercolani and Castelli, 2017), the assimilation of remotely sensed datasets, such as SM, is a more recent development as this source of data has become available only over the last decades (e.g., Parada and Liang, 2004; De Lannoy et al., 2007; Jia et al., 2009; Matgen et al., 2012; Su et al., 2013b; Chen et al., 2014; Mohanty et al., 2017).

SM is a key variable in hydrological models. In many of them, including Variable Infiltration Capacity (VIC, Liang et al.,
1994), "Hydrologiska Byråns Vattenbalansavdelning" (HBV, Bergström, S., 1976), "modèle du Génie Rural à 4 paramètres Journalier" (GR4J, Perrin et al., 2003), etc., SM controls the partitioning of water and energy fluxes. Hence, improving its representation within a numerical model has the potential of improving predictions of the key hydrological variables. In this context, SM data derived from various satellite missions such as ASCAT (e.g., Brocca et al., 2010, 2012; Dharssi et al., 2011;

Draper et al., 2011) and AMSR-E (e.g., Reichle et al., 2007; Yang et al., 2007; Draper et al., 2009) have been assimilated into land surface or hydrological models (e.g., Draper et al., 2012; Renzullo et al., 2014).

Since November 2009, the passive Microwave Imaging Radiometer with Aperture Synthesis (MIRAS) onboard the Soil Moisture and Ocean Salinity (SMOS) satellite has been observing top-of-the-atmosphere brightness temperature (Tb). The MIRAS sensor is sensitive to 1.4 GHz (L-band) emissions and takes multi-angular measurements at vertical and horizontal polarisations (Kerr et al., 2001). The algorithm used for the retrieval of SM values from SMOS Tb is based on numerical modelling (Kerr et al., 2012). In past studies, SM estimates retrieved from SMOS Tb were most of the time assimilated into land surface and sometimes in conceptual hydrological models (Wanders et al., 2014; Lü et al., 2016). However, the land surface model used for the SM retrieval and the model used for the background simulation are often different for example in terms of process representation, model structure and model forcing datasets (e.g., air and soil temperature) (De Lannoy and Reichle, 2016a). In the event the background simulation is carried out using a conceptual hydrological model, these differences may be even more important, especially in terms of process representation. This potentially result in inconsistencies in the way SM is simulated by the model and retrieved from the observation. Moreover, De Lannoy and Reichle (2016b) argued that this issue can lead to correlation between retrieved and simulated SM errors that cannot be easily handled by data assimilation filters. As a consequence, recent studies (e.g., De Lannoy and Reichle, 2016a; Lievens et al., 2016; Rains et al., 2017, 2018; Munõz-Sabater et al., 2019) have aimed to directly assimilate SMOS Tb into such land surface models. To do so, these studies used as observation operator of the assimilation filter a radiative transfer model (e.g., the Community Microwave Emission Modelling platform (CMEM), de Rosnay et al., 2009) that allows to derive Tb from SM simulations. In this context, De Lannoy and Reichle (2016a) showed that assimilating either SM retrievals or observed Tb yields almost the same correlation level between *in situ*-observed and simulated SM (average correlation equals 0.6 based on the records obtained from many measurement sites distributed across the United States of America) and Lievens et al. (2016) showed that the assimilation of SM retrievals slightly outperforms the assimilation of observed Tb.

Currently, for applications at the large scale, there is a tendency to rely on more complex physically-based hydrological models in order to better capture the hydrological processes at hand (Devia et al., 2015). However, this may be sometimes detrimental to large-scale operational hydrology, due to the increased computational demand and the potential unavailability of the required datasets for parameter estimation. Faster models are key tools for carrying simulations at a large scale without implying a high computational demand. Faster models are therefore powerful for near real-time forecasting applications and when large ensemble of model runs are required. In this context, conceptual models that allow for more efficient and rapid simulations offer an alternative to more physically-based land surface models (Devia et al., 2015; El Hassan et al., 2013). The main argument against the use of a conceptual model is often the need for site-specific parameter calibration that is often infeasible in data scarce areas. However, with the recent increase of satellite missions providing global observations of key hydrological variables at high temporal and spatial resolution, it becomes possible to envisage the calibration of conceptual models even at the large scale. Hence, a science question that is worth investigating is whether a flexible conceptual model, relying on parameter calibration, can reach the performance level of a more complex physically-based model for soil moisture simulations at large scales.

The SUPERFLEX modelling framework (Fenicia et al., 2016) enables tailoring the model structure (i.e., adapt the model architecture via reorganising constituting reservoirs) for the specific needs of the application. In particular, here we seek for a simplified representation of the main controlling processes, and computational efficiency in order to perform rapid simulations over large areas and for long periods. Compared to more physically based land surface models, the model built with SUPER-FLEX offers fast running simulations without the need for high performance computing facilities and allows for adapting the model spatial resolution and soil stratification to the characteristics of the satellite datasets that are to be assimilated.

Following the study by Rains et al. (2017), we evaluate here the potential of SMOS Tb assimilation for improving SM simulations of this distributed conceptual hydrological model. The general objective of this study is to assess the performance of a soil moisture prediction chain based on the assimilation of SMOS Tb into a coupled SUPERFLEX-CMEM model. Moreover, we propose to compare it to the one developed in Rains et al. (2017) based on the Community Land Model (CLM, Oleson et al., 2013). To enable a fair and meaningful evaluation and comparison, we use a quasi-identical experimental set up to the one of Rains et al. (2017), except that we use here the SUPERFLEX instead of the CLM model to simulate soil moisture. As a test case, we use the Murray-Darling basin in Australia and we simulate distributed time series of soil moisture over the period 2010-2015.

The specific objectives of this study are as follows: (i) to compare the SUPERFLEX and CLM models in their ability to simulate Tb and soil moisture, and (ii) to evaluate the improvement in model predictions when assimilating SMOS Tb observations. It is worth mentioning that, here and in the remainder of the paper "'simulated Tb"' is used for naming Tb that is derived from the simulated soil moisture using the radiative transfer model parametrized to emulate SMOS observations. The simulated Tb is therefore meant to emulate SMOS observation based on simulated soil moisture. As an additional objective, we also propose to evaluate how the assimilation of SMOS Tb can help in improving evapotranspiration predictions.

In the next sections, we first present the database used for the experiment, the coupling between the hydrological (SUPER-FLEX) and the radiative transfer (CMEM) models and the data assimilation experiment. Next, we calibrate the hydrological model using SMOS Tb observations, we evaluate the forward run of the SUPERFLEX-CMEM prediction chain and we compare the performances with the ones obtained in Rains et al. (2017). Then, we assess and discuss the results of the assimilation experiment using the study by Rains et al. (2017) as a benchmark. As a further discussion element, we finally evaluate the impact of the assimilation of SMOS Tb on evapotranspiration simulations.

## 2  Material and Method

### 2.1  Study area and available data

#### 2.1.1  Study Area

The study area is the Murray-Darling Basin (MDB) in South-Western Australia. The three main rivers of the MDB, namely rivers Darling, Murray and Murrumbigee are among the longest rivers in Australia. The MDB covers an area of 1.06 million km$^2$, representing approximately 14 % of the land surface of Australia. Due to its large dimensions, the basin exhibits various

climate regimes, from sub-tropical in the north to semi-arid in the west and mostly temperate in the south. The average inter-annual rainfall ranges from up to 1,500 mm in the eastern side and less than 300 mm in the western side of the MDB (MDBA, 2018). The average inter-annual temperature ranges from ca. 10 °C in south-eastern and ca. 20 °C in western side of the MDB (MDBA, 2018).

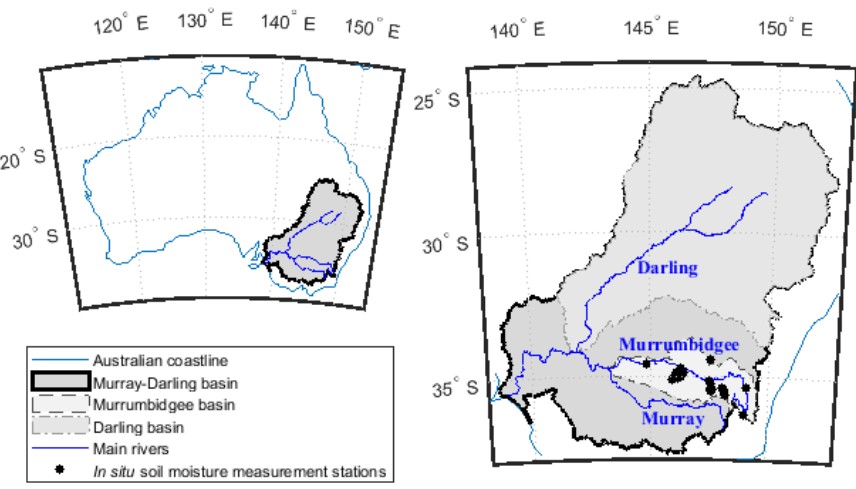

**Figure 1.** Study area: Murray-Darling, Murray, Darling and Murrumbidgee Basins, riverstream courses and *in situ* soil moisture measurement locations.

### 2.1.2    Meteorological forcings

Time series of rainfall and 2-m air and soil temperature predictions are globally available at a 3-hourly time step and 0.25° spatial resolution (downscaling from the original 0.75° spatial resolution) from the ERA-Interim reanalysis data set (Dee et al., 2011). It would have been possible as well to use the more recent and accurate ERA5 dataset, but we decided here to use ERA-Interim as it was also used in Rains et al. (2017). From this data set, for each grid cell lying within the limits of the MDB, we extracted rainfall and soil and 2-m air temperature for the period 2009 to 2016. Soil temperature was extracted for the two upper soil layers, having depths of 7 and 21 cm respectively. Next, the resulting time series were uniformly redistributed to an hourly time step. For the accumulated variable (i.e., rainfall) the predicted amount was redistributed uniformly from 6h accumulation to 1h accumulation in order to keep water balance. For the other variables (i.e., air and soil temperature), the value was imposed constant over 6h and equal to the era-interim predicted value. The potential evapotranspiration (Ep) was estimated from the air temperature data using the Hamon formula (Hamon, 1963). Rainfall and Ep time series are used as inputs of the SUPERFLEX hydrological model (see section 2.2.1). Soil and air temperature time series are used as inputs of the CMEM radiative transfer models (see section 2.2.2)

### 2.1.3 SMOS Tb observations

The SMOS database used in this study is identical to the one used in the study by Rains et al. (2017). It covers the period 2010 to 2015 and consists of SMOS Level 3 daily Tb at horizontal polarisation and 42.5° incidence angle. They are provided by the Centre Aval de Traitement des Données (CATDS) (version 310). SMOS acquisitions having probabilities of radio frequency interference (RFI) greater than 0.2, Data Quality Index higher than 0.07 or activated science flags, namely strong topography, snow, flooding, urban areas, coastal zone and precipitation were filtered out from the initial database. The filtered observation data were resampled from the Equal-Area Scalable Earth Grid 2 (EASE2) 25 km grid to a 0.25° model grid aligned with that of the ERA-interim dataset by using inverse-distance interpolation.

### 2.1.4 *In situ* soil moisture observations

As an independent dataset for evaluating the model results, we make use of *in situ* soil moisture measurements from OzNet and CosmOz measurement networks (Smith et al., 2012). These datasets provide time series of soil moisture acquired using, respectively, time-domain reflectometry (TDR) probes and cosmic-ray neutron probes. Depending on the type of probe, soil moisture observations are available for various soil depths, namely 5, 8, 30, 60 and 90 cm. The measurement stations are mainly located within the Murrumbidgee catchment as the latter was selected as one of the sites for SMOS calibration/validation campaigns (Peischl et al., 2012; Holgate et al., 2016; Su et al., 2013a). More details on the measurement techniques and the measurement network can be also found in other studies over Australia and the MDB (e.g., Holgate et al., 2016; Su et al., 2013a). It is worth mentioning that the *in situ* soil moisture dataset is provided with local or limited measurement footprints (a few hundreds of m$^2$ at maximum) whereas the hydrological model simulates average soil moisture over much larger areas (a few hundreds of km$^2$). As a consequence, the comparison between model results and *in situ* observation necessarily suffers from scale-representativeness issues.

### 2.1.5 *In situ* flux tower measurements

As an additional independent dataset for evaluating the model results, we also make use of *in situ* flux tower measurements from the TERN OzFlux measurement network (http://www.ozflux.org.au/). This dataset provides, among other variables, time series of latent heat fluxes that were converted into actual evapotranspiration rates using the latent heat of vaporization constant. The measurement stations are mainly located in the southern part of the MDB. Moreover, the *in situ* evaporation data, just like the previously described soil moisture data, are provided with local or limited measurement footprints.

## 2.2 The soil moisture and Tb prediction chain

### 2.2.1 The conceptual hydrological Model

The SUPERFLEX modelling framework (hereafter denoted SFX, Fenicia et al., 2011, 2016) is used to build the hydrological model. This modelling framework was developed with the aim to facilitate model development and allow model structure

comparisons. The modelling platform is based on generic building components that can be configured and combined in various ways to generate different model architectures. Hydrologists can therefore hypothesize, build and test different model structures. For example, it allows for adapting the model structure to the forcing and observation datasets (e.g., in terms of spatial and vertical resolutions) and specific characteristics of the catchment. In the context of this study, we take advantage of this flexibility and define the model architecture in such a way that it allows to easily ingest globally available meteorological forcing data and at the same time integrate Tb as observed by the SMOS satellite. The model is therefore distributed over grid cells of 0.25°aligned on the grid used in the ERA-Interim dataset and simulations are carried out at an hourly time step.

The architecture of the developed model is represented in Figure 2 for one model grid cell. It is mainly composed of two stratified upper root zone layers represented by two reservoirs, namely UR$u$ and UR$l$. The grey box in Figure 2 also identifies the deeper reservoirs and the routing function that simulates subsurface and surface runoff based on deeper soil layer water storage. In SFX, the deeper reservoirs are typically two interconnected fast and slow reservoirs, with associated lag functions whose outflows are summed up to compute the surface runoff (Fenicia et al., 2016). In this study, since we focus on the two upper root zone layers that are of interest for simulating soil moisture, the deeper reservoirs and the routing function are switched off and not further referred to in the remainder. It is worth mentioning that the removal of the deeper reservoirs of SFX has no effect on the soil moisture simulations as in SFX there is no upward water circulation from the deeper reservoirs to the upper ones. As a matter of fact, when deeper reservoirs are switched off, water exits root zone soil layers based on the usual equations. The soil moisture simulations within both root zone reservoirs are therefore not impacted.

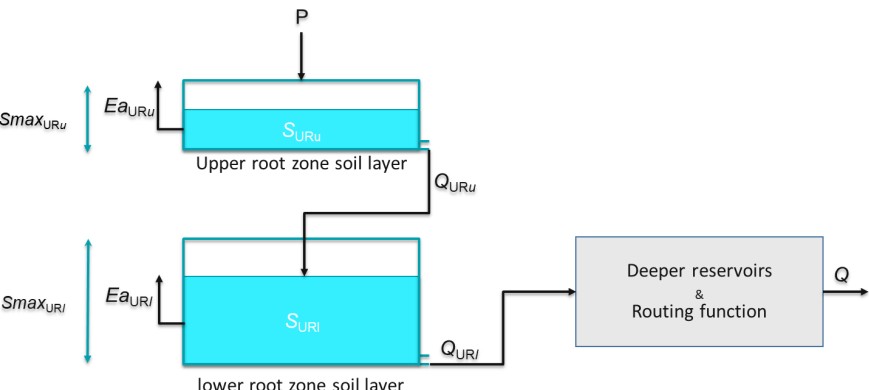

**Figure 2.** SUPERFLEX Model architecture.

The upper reservoir (UR$u$) is fed by precipitation and looses water through evapotranspiration to the atmosphere and percolation to the second reservoir (UR$l$). The latter is then fed by the incoming percolation from the first reservoir and looses water through evapotranspiration to the atmosphere and percolation to deeper soil. Outflow $Q$ from the two root zone layers is estimated based on the simulated storage $S$ and the incoming water amount using a power function with exponent $\alpha$:

$$Q_{\mathrm{UR}x,i}(t) = P_{\mathrm{UR}x,i}(t) \times \left( \frac{S_{\mathrm{UR}x,i}(t)}{Smax_{\mathrm{UR}x,i}} \right)^{\alpha_{\mathrm{UR}x,i}} \qquad (1)$$

where $t$ is time, $i$ represents the model grid cell number, $x$ stands for upper ($u$) or lower ($l$) reservoir, $P_{\mathrm{UR}x,i}$ is the input to the reservoir (precipitation for the upper reservoir, outflow from the upper reservoir for the lower reservoir), and $Smax$ is a parameter representing maximum storage capacity.

The actual evapotranspiration ($Ea$) from the two soil layers is estimated based on the simulated storage within the considered reservoir and the potential evapotranspiration ($Ep$) using a power function with exponent $\beta$:

$$Ea_{\mathrm{UR}x,i}(t) = Ep_i(t) \times \left( \frac{S_{\mathrm{UR}x,i}(t)}{Smax_{\mathrm{UR}u,i} + Smax_{\mathrm{UR}l,i}} \right)^{\beta_{\mathrm{UR}x,i}} \tag{2}$$

The variation of storage within the two reservoirs is estimated by solving the water balance equation:

$$\frac{\mathrm{d}S_{\mathrm{UR}x,i}(t)}{\mathrm{dt}} = P_{\mathrm{UR}x,i}(t) - Q_{\mathrm{UR}x,i}(t) - Ea_{\mathrm{UR}x,i}(t) \tag{3}$$

For each reservoir, the soil moisture is derived from the storage according to:

$$\theta_{\mathrm{UR}x,i}(t) = C_{\mathrm{EF}x,i} \frac{S_{\mathrm{UR}x,i}(t)}{Smax_{\mathrm{UR}x,i}} \tag{4}$$

where $\theta$ is the predicted soil moisture and $C_{\mathrm{EF}}$ a so-called effective field capacity.

In the model architecture, the two root zone reservoirs are meant to conceptually represent two stratified soil layers allowing to simulate soil moisture over different soil depths. To maintain constant depths of these two layers over the model domain (namely 7 and 21 cm in accordance to the depth of the two upper soil layers depicted in the ERA-interim dataset) the respective reservoir maximum capacities are computed depending on the $C_{\mathrm{EF}}$ considering that the maximum storage capacity of a soil layer can be derived from the $C_{\mathrm{EF}}$ and the soil layer depth $d$ according to:

$$Smax_{\mathrm{UR}x,i} = C_{\mathrm{EF}x,i} \times d_{\mathrm{EF}x} \tag{5}$$

It is worth noting here that the model structure presented beforehand is replicated on each model grid cell $i$. As a matter of fact, the distributed SFX implemented in this study does not simulate lateral flows within the root zone soil layers. As a consequence, for each grid cell, the model has six calibration parameters, namely $C_{\mathrm{EF}u,i}$, $\alpha_{\mathrm{UR}u,i}$, $\beta_{\mathrm{UR}u,i}$, $C_{\mathrm{EF}l,i}$, $\alpha_{\mathrm{UR}l,i}$, $\beta_{\mathrm{UR}l,i}$. Moreover, the maximum storage capacities $Smax_{\mathrm{UR}x,i}$ are computed based on fixed soil layer depths $d_{\mathrm{EF}x}$ and calibrated effective field capacities $C_{\mathrm{EF}u,i}$.

From the SFX model description, it appears that this conceptual model is much simpler than CLM in particular for the following reasons:

– Whereas CLM estimates latent heat fluxes between soil and atmosphere based on aerodynamic diffusion equation and Monin-Obukhov similarity theory (Oleson et al., 2013), taking into account many information such as soil and vegetation type, surface roughness, atmospheric stability, the vegetation coverage, SUPERFLEX lumps energy balance contribution to water balance via a simpler potential evapotranspiration formula, namely the Hamon formula.

– In this study, SFX is structured with two soil layers (respectively 0-7cm and 7-21cm) whereas CLM considers five layers for the same soil depth range.

**Table 1.** Main processes implemented in the CLM- and SFX-based studies.

| Main Processes | CLM | SFX |
|---|---|---|
| Surface energy fluxes | Air and soil heat fluxes (Monin-Obukhov similarity) | |
| Evaporation | Latent heat flux (Monin-Obukhov similarity) | Lumped evapotranspiration (Hamon formula) |
| Transpiration | Phenology and plant development status | |
| Subsurface vertical flow | Across 10 soil layers (adapted Richards equation) | Across 2 soil layers (power low dynamics) |
| Subsurface lateral Flow | Only in the saturated groundwater | - |

– The set up of CLM therefore requires much more input data (e.g., soil types and land use), and Superflex has a limited number of parameters (6 parameters per cell) compared to CLM (potentially tens of parameters per cell, Hou et al., 2012).

It is worth noting here that this list is not exhaustive and that many other processes are further simplified in SFX compared to CLM. A detailed presentation of CLM is available in Oleson et al. (2013). Table 1 reports the main processes represented in our implementation of SFX and in the CLM set up of (Rains et al., 2017).

### 2.2.2 The radiative transfer model

To simulate Tb using soil moisture predictions of the SFX hydrological model, we use the Community Microwave Emission Model version 5.1 (CMEM, de Rosnay et al., 2009). The parametrization of CMEM and most of the forcings (except SM and soil temperature) are identical to the ones used in the study of Rains et al. (2017) in order to enable a meaningful comparison between both experiments. In particular, the time invariant input data (i.e., soil sand and clay fractions, permanent water surface fractions, ground elevation and vegetation cover types) as well as the equations used to run CMEM are exactly the same. The ECOCLIMAP vegetation classes (Champeaux et al., 2005) are used to provide CMEM with the plant functional types. The development cycle of vegetation classes is defined in CMEM based on the leaf area index (LAI) (Rains et al., 2017). LAI is interpolated at daily scale from a monthly dataset for low vegetation and a constant LAI value is fixed for high vegetation. The dielectric constant computation is carried out using the Mironov model (Mironov et al., 2004) and the required effective temperature is computed via the Wigneron model (Wigneron et al., 2001). The Fresnel, Choudhury (Choudhury et al., 1979) and Wigneron (Wigneron et al., 2007) models are used for assessing smooth surface emissivity, soil roughness and vegetation opacity respectively. Atmospheric contributions are estimated via the Pellarin method (Pellarin et al., 2003). However, the soil layer depths in CMEM are identical to the ones used in the SFX model and the soil moisture simulated by SFX is used as input of CMEM. Moreover, as SFX does not integrate energy balance processes (while CLM does), the soil temperature is in our experiment derived from the ERA-Interim dataset.

### 2.2.3 Model Calibration

On each grid cell, the SFX model has 6 calibration parameters (see section 2.2.1). To carry out the calibration, Monte-Carlo simulations using latin hypercube sampling within plausible parameter ranges are carried out. To do so, parameter sets are first randomly generated within such plausible parameter ranges. Next, a SFX-CMEM simulation is carried out for each individual parameter set and the simulated Tb is compared, at the grid cell scale, to the values derived from SMOS observations. Eventually, for each individual model grid cell, the parameter set yielding the lowest unbiased root mean square deviation ($ubRMSD$, Entekhabi et al., 2010, see Eq. 6) while comparing simulated and SMOS-derived Tb is selected as optimal. The $ubRMSD$ is chosen here has it allows to remove the bias between simulated and observed soil moisture (and Tb) which is common in brightness temperature assimilation studies.

$$ubRMSD = \sqrt{\left\langle \left( (y^s(t) - \langle y^s \rangle_t) - (y^o(t) - \langle y^o \rangle_t) \right)^2 \right\rangle_t} \qquad (6)$$

where $y^s$ is the simulated Tb and $y^o$ the observed one and $\langle . \rangle_t$ indicates the average over time. The parameters of the CMEM model were not adjusted and their default values were used to keep the current experiment quasi identical to one of Rains et al. (2017).

## 2.3 Data Assimilation

### 2.3.1 Data assimilation filter

In this section, we present the method used for assimilating SMOS Tb into the SFX-CMEM coupled models. The method proposed in this study uses as assimilation filter a Local Ensemble Transform Kalman Filter (LETKF) introduced by Hunt et al. (2007) and implemented by Miyoshi and Yamane (2007). As usual in ensemble Kalman filtering, the uncertainty in model predictions is represented via a set of $k$ stochastic model realizations ($k$=32 in this experiment) having different perturbed forcings and/or parameters while the model and observation errors are assumed to be normally distributed. The localisation is set up so that the assimilation is carried out at the model grid scale. The observation operator is linearly approximated during the analysis step in the LETKF (see Eq.18 in Hunt et al., 2007). As argued in Hunt et al. (2007), LETKF is deterministic as no additional random error is added to the observation. Let us denote our non-linear model $\mathcal{M}$, namely SFX, that propagates state variables in time (including soil moisture $\theta$) between two assimilation time steps (Eq. 7).

$$x_{n,j}^{b} = \mathcal{M}\left(x_{n-1,j}^{a}\right) \qquad (7)$$

where $x_{n,j}^{b}$ is the background at time $t_n$ when the assimilation is supposed to be carried out for ensemble member number $j$, and $x_{n-1,j}^{a}$ is the analysis computed at time $t_{n-1}$, i.e. the previous assimilation time step.

The step prior to the assimilation is to run the hydrological model between $t_{n-1}$ and $t_n$ to yield the background ensemble. In our study, the application of the LETKF proposed by Hunt et al. (2007) consists of the seven main steps listed hereafter (please note that the temporal index $n$ is not repeated later on for the sake of conciseness).

1. Apply the observation operator, namely CMEM, to the model background ensemble to form the observational background ensemble $y^{\mathrm{b}} = [y_1^{\mathrm{b}}, ..., y_k^{\mathrm{b}}]$.

2. Compute the ensemble observational background perturbations based on the ensemble mean $\overline{y^b}$:

$$\boldsymbol{Y}^{\mathrm{b}} = [y_1^{\mathrm{b}} - \overline{y^{\mathrm{b}}}, ..., y_k^{\mathrm{b}} - \overline{y^{\mathrm{b}}}]$$

3. Compute the ensemble background perturbations based on ensemble mean $\overline{x^b}$:

$$\mathbf{X}^{\mathrm{b}} = [x_1^{\mathrm{b}} - \overline{x^{\mathrm{b}}}, ..., x_k^{\mathrm{b}} - \overline{x^{\mathrm{b}}}]$$

4. Compute the matrix $\tilde{\mathbf{P}}^a = \left[ (k-1)\mathbf{I} + \left(\mathbf{Y}^{\mathrm{b}}\right)^T \mathbf{R}^{-1} \mathbf{Y}^{\mathrm{b}} \right]^{-1}$ where $\mathbf{I}$ is the identity matrix and $\mathbf{R}$ the observation error covariance matrix.

5. Compute the matrix $\mathbf{W}^{\mathrm{a}} = \left[ (k-1)\tilde{\mathbf{P}}^{\mathrm{a}} \right]^{1/2}$

6. Compute the k-dimensionnal vector $\overline{w}^{\mathrm{a}} = \tilde{\mathbf{P}}^{\mathrm{a}} \left(\mathbf{Y}^{\mathrm{b}}\right)^T \mathbf{R}^{-1} \left( y^o - \overline{y}^{\mathrm{b}} \right)$ and derive $w_j^{\mathrm{a}}$ by adding $\overline{w}^{\mathrm{a}}$ to each column of $\mathbf{W}^{\mathrm{a}}$

7. Compute the individual member analysis $x_j^{\mathrm{a}} = \overline{x}^{\mathrm{b}} + \mathbf{X}^{\mathrm{b}} \mathbf{w}_j^{\mathrm{a}}$

This process is repeated for each cell of the model domain where a SMOS observation is available at time step $t_n$. Once the analysis has been carried out, state variables, namely the storage in the two soil layers of SFX (section 2.3.1), are updated and the simulation is resumed until the next assimilation time step.

As mentioned in many studies dealing with the assimilation of satellite SM or Tb (e.g. Al Bitar et al., 2012; Matgen et al., 2012; Rains et al., 2017; Al-Yaari et al., 2017), bias removal prior to the assimilation is often a necessary step. In our study, we reduce the bias between simulations and observations by deriving model and observation anomalies, following an identical approach to the one described in Rains et al. (2017). Anomalies are defined as the difference between the original Tb time series and their inter-annual climatologies (time-average SMOS observation acquired in a 20 days-sliding window centred on the considered day of year). The climatology is computed by first smoothing the Tb time series using a 20-days moving average and next computing the inter-annual average of the smoothed signal. The model background used in the assimilation filter is the simulated Tb anomaly computed as the difference between the simulated Tb and its climatology, computed as the climatology of the ensemble mean of the open loop run. The data assimilation is therefore carried out based on the simulated and observed Tb anomalies.

### 2.3.2 Ensemble generation

To generate an ensemble of simulated Tb, the meteorological forcings of the SFX-CMEM models derived from the ERA-interim dataset, namely the rainfall and the air and soil temperature time series are randomly perturbed. As in Rains et al. (2017), the perturbation applied to rainfall time series is multiplicative and randomly generated from a log-normal statistical

**Table 2.** Similarities and differences between the two quasi-identical assimilation experiments.

| Experiment characteristics | CLM-based study | SFX-based study |
| --- | --- | --- |
| Number of root zone soil layers | 5 | 2 |
| Ensemble generation | Perturbed input rainfall<br>Perturbed input air temperature<br>Perturbed soil texture parameters | Perturbed input rainfall<br>Perturbed input air temperature<br>- |
| Simulation time steps | Identical: 1 hour | |
| Model grid size | Identical: 0.25° | |
| Source of model forcings | Identical: ERA-interim | |
| Radiative transfer model | Identical: CMEM using the same parametrization | |
| Assimilation filter | Identical: LetKF | |
| Assimilated satellite observations | Identical: SMOS Tb | |

distribution of mean 0 and standard deviation 0.5. The air temperature time series are perturbed using an additive Gaussian random noise of mean 0 K and standard deviation 2.5 K. Each time step and each model grid cell has an independently drawn random perturbation. Moreover, to maintain a set up similar to the one used in Rains et al. (2017) where air temperature perturbations are propagated to the soil temperature via the CLM model, perturbed soil temperature predictions are here drawn
from the perturbed air temperature. This is done in two steps. First, linear regressions are carried out on each grid cell between the ERA-interim predictions of air temperature and soil temperature (separately for the two soil layers). Next, perturbed soil temperatures are derived from the perturbed air temperatures based on the coefficients obtained from the linear regressions. This allows to maintain a certain level of consistence between perturbed air and soil temperatures for each ensemble member. The main difference between our experiment and the one of Rains et al. (2017) is that we do not perturb soil texture as this
parameter of CLM does not apply to SFX. Table 2 reports the similarities and differences between the two quasi-identical assimilation experiments.

## 2.4  Analyses used to evaluate the proposed soil moisture prediction chain

The proposed modelling framework is evaluated and compared to the one proposed in Rains et al. (2017) using a series of empirical tests:

1. We assess the performance of the calibrated conceptual SFX model by comparing, via the Pearson's correlation (refered to as $\rho$ in the remainder), the ubRMSD (Eq. 6) and the mean bias, the simulated and observed SMOS Tb.

2. We compare SFX-based model performance to the one of the forward CLM model previously introduced in Rains et al. (2017). To do so, we make use of the root mean square deviation (RMSD) and the $\rho$ together with Taylor diagrams computed based on the comparison between CLM (resp. SFX) model simulations and observations of Tb (SMOS observation) and SM (*in situ* measurements).

3. We assess the effect of the assimilation of SMOS Tb by comparing the open loop and the assimilation simulations of SM with *in situ* SM measurements, via the RMSD, the $\rho$, the ubRMSD, the assimilation efficiency (Eq. 8) and Taylor diagrams and we analyse the spatial distribution of $\rho$ improvement by mapping the $\rho$ changes between predictions and in situ measurements of soil moisture at each stations.

4. We further evaluate the influence of the assimilation of SMOS observations on the prediction of evapotranspiration by comparing the open loop and the assimilation simulation of evapotranspiration with *in situ* measurements. This comparison is carried out graphically via Taylor Diagrams and time series plot and numerically via the percentage improvement (Eq. 9).

The assimilation efficiency is computed as follows:

$$
E(t_n) = \begin{cases} \left(1 - \frac{SD_{An}(t_n)}{SD_{OL}(t_n)}\right) * 100, & \text{if } SD_{An} \leq SD_{OL} \\ \left(\frac{SD_{OL}(t_n)}{SD_{An}(t_n)} - 1\right) * 100, & \text{if } SD_{An} > SD_{OL} \end{cases}
\tag{8}
$$

$$
\text{With} \quad \begin{cases} SD_{OL}(t_n) = \sum_{t=t_n}^{t_{n+1}-1} (\theta_{OL}(t) - \theta_{Obs}(t))^2 \\ SD_{An}(t_n) = \sum_{t=t_n}^{t_{n+1}-1} (\theta_{An}(t) - \theta_{Obs}(t))^2 \end{cases}
$$

where $E(t_n)$ is the efficiency of the analysis at time step $t_n$, $SD_{An}$ the squared deviation of the analysis run, $SD_{OL}$ the squared deviation of the open loop run, $\theta_{Obs}$ the observed soil moisture, $\theta_{An}$ the analysis soil moisture prediction and $\theta_{OL}$ the open loop soil moisture prediction. The efficiency evaluates the squared deviation change as a result of the assimilation. Positive values indicate an error reduction, while negative ones indicate that the squared deviation increased after the analysis step.

The percentage improvement is computed as follows:

$$
Imp(t) = \frac{\|Ea_{OL}(t) - Ea_{Obs}(t)\| - \|Ea_{An}(t) - Ea_{Obs}(t)\|}{\|Ea_{OL}(t) - Ea_{Obs}(t)\|} * 100
\tag{9}
$$

where $t$ is the time step, $Imp$ the percentage improvement and $Ea_{Obs}$, $Ea_{OL}$ and $Ea_{An}$ respectively the observed, background and analysis evapotranspiration. The positive (resp. negative) percentage improvement values indicate that absolute errors are reduced (resp. increased) as a result of the assimilation of SMOS Tb.

## 3 Results and discussion

In this section, the performance of the conceptual SFX model is assessed and compared to the one of the physically-based CLM land surface model by comparing simulated and observed time series of Tb and soil moisture.

## 3.1 Evaluation of the calibrated SUPERFLEX hydrological model

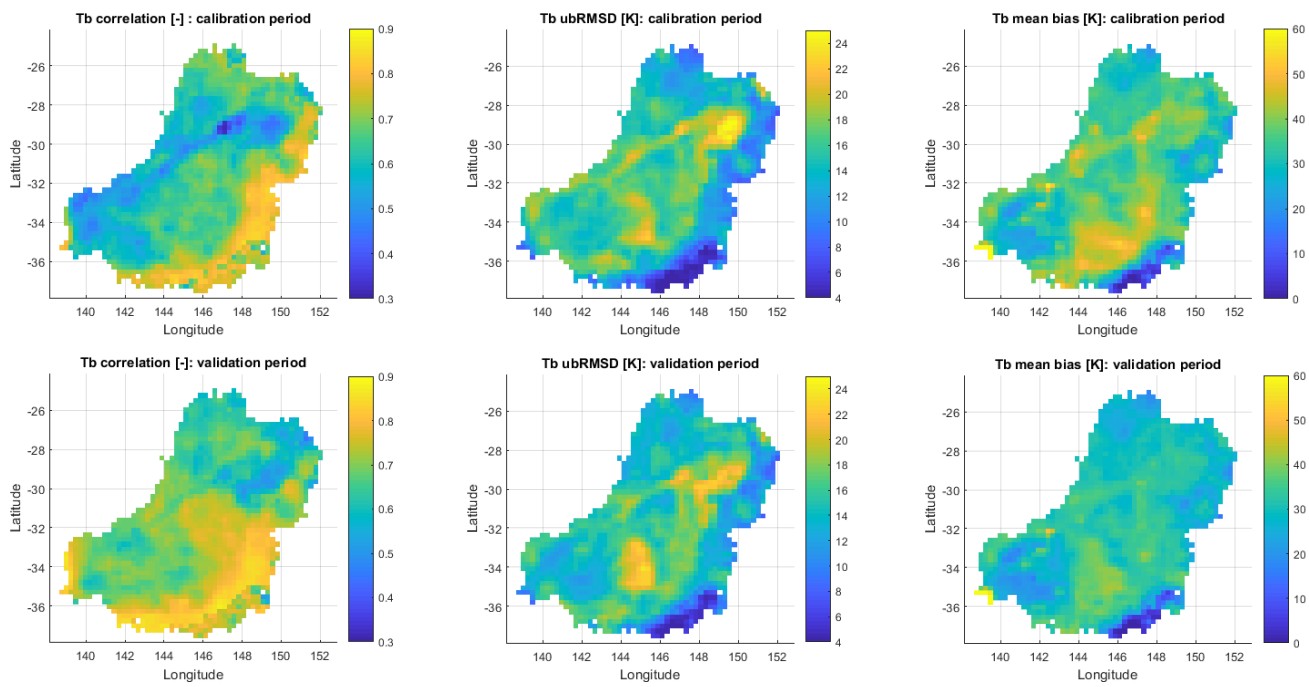

**Figure 3.** Performance and error metrics of Superflex simulated Tb using as a reference SMOS Tb: $\rho$ coefficients (left hand side panels), unbiased root mean square deviations (ubRMSD, center panels) and Mean Biases (right hand side panels) between SFX-CMEM predictions and SMOS observations of Tb during the calibration (top panels) and the validation (bottom panels) periods.

Figure 3 shows the $\rho$ coefficient, the RMSD and the Mean Bias maps obtained by comparing SFX-simulated and SMOS-observed Tb time series and Table 3 reports the associated spatial statistics during the calibration (2010-2011) and the validation (2012-2015). The p-values associated to $\rho$ coefficients are all below 0.01 lending therefore weight to the significance of the $\rho$ between simulated Tb and SMOS Tb. From this figure and this table, the following results can be noted:

1. The calibrated model yields rather satisfying predictions of Tb. In addition, the obtained performances are comparable to those obtained in Rains et al. (2017). In particular, in our study we have an average $\rho$ of 0.7, an average ubRMSD of 14.8 K and an average bias of 30.21 K during the validation period. In the study of Rains et al. (2017) using CLM, the RMSD has an average value of 30 K and the average $\rho$ a value 0.7.

2. The three performance metrics have rather similar values and spatial variability when computed during the calibration and the validation periods although slight differences are visible in Table 3.

3. A general gradient in the performance of SFX can be seen from the eastern to the western part of the basin whereas this gradient is not observed in the CLM.

4. The lowest performances are mainly exhibited on pixels located in the Darling river floodplain (Figure 1).

Results 1 and 2 leads us to conclude that model results are satisfactory in view of previous applications. Result 3 can be explained based on the fact that the hydrological regimes vary from east to west in the MDB. Whereas the eastern part is more dominated by rainfall, the western part receives limited amounts of rainfall and evapotranspiration plays a more important role in the hydrological cycle. Considering that in our set up, the representation of the evapotranspiration is rather simplistic as it is based on the Hamon formula, this could explain the poorer performance of the model in the western part of the basin. Indeed, the simplified representation of SFX model does not allow to adequately capture the evapotranspiration-induced controls of soil moisture, which revealed poorer performance of the model in the western part of the basin. Result 4 can be explained considering that the input data used for running CMEM concerning the fraction of the grid cell covered by surface water. This input is considered invariant over time in our set up, while in reality an important number of lakes and ponds of the Darling river floodplain are periodically drying and filling up during the year, potentially modifying the water fraction on the corresponding model grid cells.

**Table 3.** Spatial statistics of simulated Tb performance metrics (computed using as reference SMOS Tb). Cal: during Calibration Period. Val: during Validation Period

| Perfomance metric Spatial statistic | $\rho$ [-] | | ubRMSD [K] | | Mean Bias [K] | |
|---|---|---|---|---|---|---|
| | Cal | Val | Cal | Val | Cal | Val |
| Mean | 0.65 | 0.70 | 14.75 | 14.8 | 34.70 | 30.21 |
| Median | 0.65 | 0.69 | 15.31 | 14.86 | 35.37 | 31.02 |
| Mode | 0.32 | 0.44 | 3.91 | 4.25 | 0.10 | -1.11 |
| Skewness | -0.14 | -0.08 | -0.61 | -0.40 | -0.01 | 0.75 |
| Kurtosis | 2.78 | 2.78 | 3.67 | 3.98 | 12.06 | 23.54 |

## 3.2 Comparison of the performances of the SUPERFLEX and CLM models

To compare the SFX-based model performance to the one of the forward CLM model previously introduced in Rains et al. (2017), we first make use of Taylor diagrams (Figure 4). These represent useful tools to evaluate and inter-compare model performances as they display on a unique plot three key performance statistics, namely the normalized standard deviation of model results, the RMSD and the $\rho$ between model predictions and observations. The normalisation of standard deviation and RMSD is carried out with respect to observed time series statistics. The perfect model would therefore be a point located in the black circle in Figure 4 with values of normalized standard deviation, normalized RMSD and $\rho$ equal to 1, 0 and 1 respectively.

Figure 4 assumes that the SMOS Tb observations are reliable and accurate. The panel on the left hand side of Figure 4 shows the spatially averaged model statistics of both models during the calibration and the validation periods. As can be

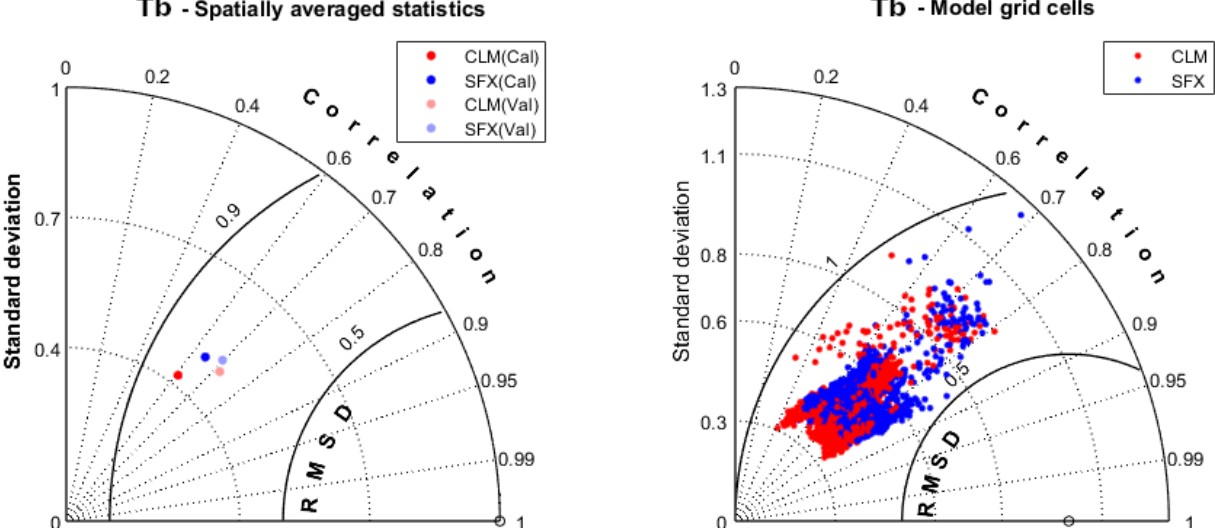

**Figure 4.** Taylor Diagrams (TD) based on the comparison between SMOS-observed and SFX- and CLM-simulated Tb respectively: TD drawn by spatially averaging model grid cell statistics, separating the calibration (cal) and validation (val) periods (left panel) and TD drawn using model grid cell statistics individually for calibration and validation periods together (right panel). Red dots indicate CLM statistics and blue dots SFX statistics.

seen, the performances of both models are similar with average $\rho$ ranging between 0.62 and 0.72 during the calibration and validation periods. Whereas SFX slightly outperforms CLM during the calibration period, CLM exhibits relatively better $\rho$ during the validation period. Overall, both models yield very similar levels of satisfying performances. The performances obtained here are as well rather similar to the ones showed in De Lannoy and Reichle (2016a) with $\rho$ between *in situ*-observed

and simulated SM of 0.6 on average over many stations located in the United States of America. One can notice that both models have a tendency to underestimate the observed variance of Tb as normalized standard deviation values are lower than 1. Our interpretation is that the two models are unable to reproduce the variance of SMOS observations mainly due to some limitations of the radiative transfer modelling (e.g.: inaccurate estimates of surface roughness or vegetation optical depth). Indeed, even with completely dry or wet soils, the simulated Tb does not reach the extreme values of the SMOS Tb. The panel

on the right hand side of Figure 4 shows the model performance for each individual model grid cell. The model statistics are here computed over the complete simulation period (calibration and validation periods). At the model grid cell scale, both model statistics cover a rather wide range of performance levels. This highlights the fact that both models, albeit yielding good overall levels of performance are less accurate for a few cells. Overall the performance metrics shown in Figure 4 confirm that the two models reach similar levels of performance.

Figure 5 shows the maps of difference in $\rho$ and RMSD between both models, as well as the map of average hourly rainfall and Ep. The top left hand side panel highlights a gradient in the $\rho$ values from west to east: the SFX-predicted Tb is better

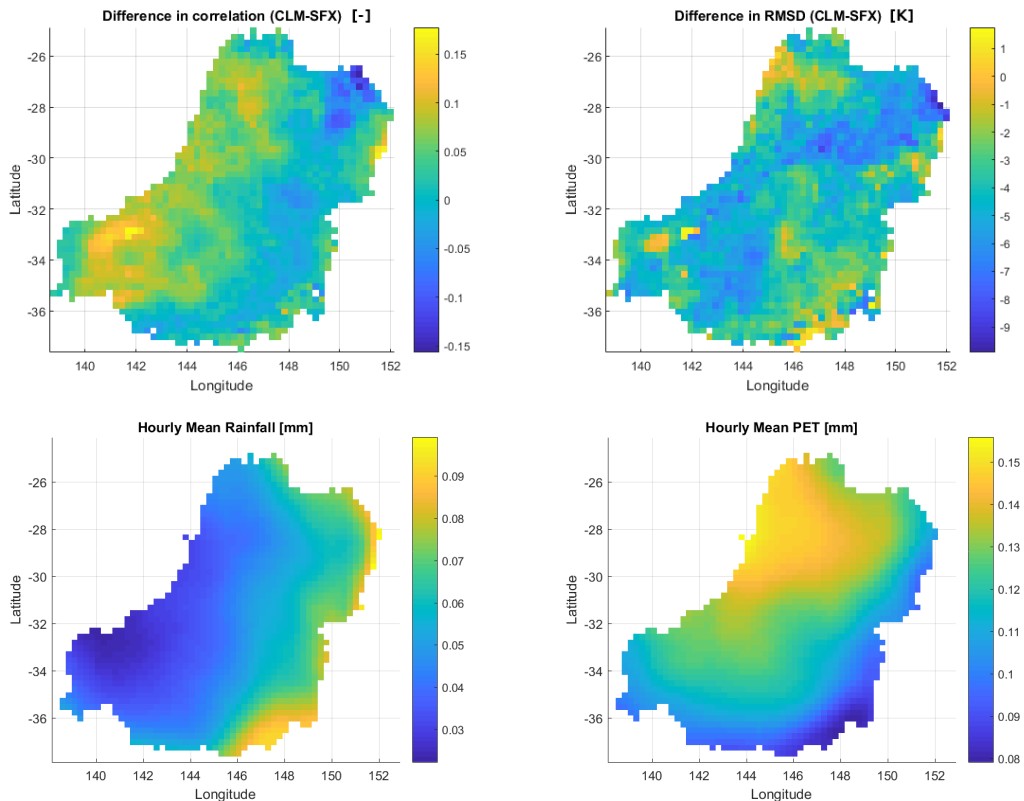

**Figure 5.** Maps of differences in $\rho$ (top left panel) and RMSD (top right panel) between CLM- and SFX-simulated Tb using as a reference SMOS observation, and maps of hourly average of input Rainfall (bottom left panel) and Ep (bottom right panel).

correlated with SMOS observations in the Eastern part of the basin where precipitation is mainly controlling soil moisture dynamics (Figure 5, bottom left panel) and the CLM-predicted Tb is better correlated with SMOS observation in the Western part, where evapotranspiration has a higher impact on soil moisture variations (Figure 5, bottom right panel). This is arguably explained by a better representation of the evapotranspiration process in CLM and a better capability of SFX to simulate fast
5  transfer of rainfall to deeper soil layers at saturation. The top right hand side panel shows a generally higher deviation in SFX-based predictions of Tb.

As SMOS observations likely suffer from significant uncertainties, we propose to further evaluate the model results using *in situ* observed soil moisture time series from a limited number of available measurement sites (Figure 1). In this context, Figure 6 shows the Taylor diagrams drawn from the comparison between time series of soil moisture observed and simulated
10  by both models for the thin upper soil layer (panel on the left hand side) and a deeper soil layer (panel on the right hand side). For the thin upper soil layer (0-8 cm depth) the observations are directly compared with the soil moisture simulations of the upper SFX reservoir. For the deeper soil layer (0-30 cm depth), the observations are compared with the average soil moisture predictions computed as the weighted mean, with weighting proportional to the maximum storage capacity of the upper and

lower SFX reservoirs (see Figure 2). Both models exhibit similar $\rho$ values. For the upper soil layer, SFX is better in capturing the observation variance. Regarding the RMSD, CLM slightly outperforms SFX with sometimes lower values for the upper soil layer. For the deeper soil layer, both models yield again similar performance levels with satisfying $\rho$ values, SFX slightly overperforming CLM.

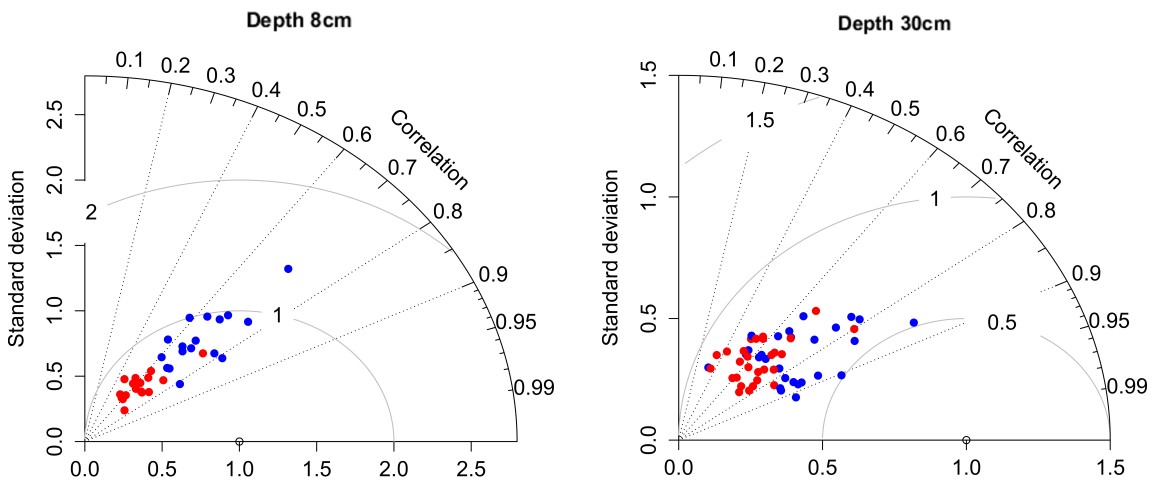

**Figure 6.** Taylor Diagrams (TD) drawn from the comparison between *in situ*-observed and SFX-simulated soil moisture for two different soil depth and the two different models. Blue dots indicate SFX statistics and red dots CLM statistics.

As a conclusion on the comparison between the forward run of both models, it can be highlighted that the two models finally reach similar performance levels when using as a reference either observed SMOS Tb or *in situ* measured soil moisture. It is also important to keep in mind that similar performance levels have been attained provided that the SFX model was calibrated whereas CLM was not calibrated using SMOS data. We argue that while a conceptual model such as SUPEFLEX requires a calibration effort because its parameter values cannot be set a priori, CLM is not supposed to be calibrated as it is physically

based and its parameters are usually derived from various input data describing for example the characteristics of the catchment (Hou et al., 2012). Moreover, one can argue that, because of a large number of parameters, calibrating CLM using SMOS data would not be an easy task especially due to the computational demand (Karagiannis et al., 2019) over a large basin such as the Murray Darling. The calibration of many parameters would consequently lead to a widely reported equifinality issue. Of course it is worth mentioning here that CLM performs satisfactorily even without any calibration.

**3.3   Effect of the assimilation of SMOS Tb on the SFX hydrological model**

The data assimilation framework proposed in section 2.3 is applied over the period 2010-2015. Each time a SMOS observation is available over a model grid cell, the assimilation filter is applied on the background and the soil water storage variables of SFX are updated. We assimilated SMOS anomalies and the error covariance of the SMOS observation anomalies $R$ is assumed constant and equal to 25 K$^2$ (as in Rains et al. (2017)). Table 4 reports the spatially averaged performance metrics of the

open loop (i.e., without assimilation) and the analysis SM simulations for two soil layer depths. As some model grid cells include several soil moisture measurement stations and with the objective to compensate for the limited footprint, the average performance metrics in Table 4 are computed both over the individual soil moisture measurement stations and over the cells where *in situ* observations are available. In the second case, all soil moisture observations available in a given model grid cell are first averaged. The performance metrics are next computed using as a reference the "averaged" observations. Eventually, the average metrics are obtained by spatially averaging the model grid-cell based metrics. As can be seen in Table 4, the assimilation allows for a moderate increase in $\rho$ for the two soil layers depicted in the model when comparing observed and simulated soil moisture time series. Specifically, the $\rho$ increases on average by more than 0.03 for both soil layer depths. These improvements are similar, although slightly lower to those obtained in the study by Rains et al. (2017, experiments DA2 and DA0), namely ca. 0.06 for upper layers and 0.03 for deeper layers. One possible explanation for the slightly lower improvements in $\rho$ for the top-layer can be found in the SFX open-loop performance being already higher ($\rho$=0.77) than that of CLM ($\rho$=0.61). This arguably limits the room for improvement as a result of the assimilation as the SFX-based open-loop outperforms the one based on CLM. Moreover, the fact that the SFX model was calibrated using SMOS Tb and forced using the ERA-interim dataset can also explain the fact that the improvement of the soil moisture predictions are slightly lower than in other studies relying on uncalibrated land surface models (e.g., Rains et al., 2017; De Lannoy and Reichle, 2016a).

**Table 4.** Time-space average values of background and analysis performance (comparison with *in situ*-observed soil moisture).

| Space averaging | Layer depth | | $\rho$ | RMSD | ubRMSD |
|---|---|---|---|---|---|
| Over measurement stations | 8cm | open loop | 0.776 | 0.1 | 0.98 |
| | | analysis | 0.801 | 0.1 | 0.98 |
| | 30cm | open loop | 0.695 | 0.11 | 0.071 |
| | | analysis | 0.727 | 0.11 | 0.072 |
| Over model grid cells | 8cm | open loop | 0.771 | 0.1 | 0.099 |
| | | analysis | 0.803 | 0.1 | 0.1 |
| | 30cm | background | 0.695 | 0.11 | 0.062 |
| | | analysis | 0.726 | 0.11 | 0.064 |

To assess the significance of the $\rho$ improvement as a result of the assimilation, we carried out Williams' significance tests (Williams, 1959). The null hypothesis in this test is that the $\rho$ improvement is not significant. For the upper soil layer the p-values are lower than 0.01 except for 2 stations where $\rho$ remains almost constant. For the deeper soil layer the p-values are lower than 0.01 except for 3 stations where $\rho$ remains almost constant or slightly decreased. This shows that the $\rho$ increase as a result of the assimilation of SMOS Tb can arguably be considered significant for the large majority of stations. To assess the significance of the $\rho$ improvement as a result of the assimilation, we carried out Williams' significance tests (Williams, 1959).

The null hypothesis in this test is that the $\rho$ improvement is not significant. For the upper soil layer the p-values are lower than 0.01 except for 2 stations where the $\rho$ values remains almost constant. For the deeper soil layer the p-values are lower than 0.01 except for 3 stations where $\rho$ remains almost constant or slightly decreased. This shows that the $\rho$ increase as a result of the assimilation of SMOS Tb can arguably be considered significant for the large majority of stations.

However, while $\rho$ values increase due to the positive effect of the assimilation, one can notice in Table 4 that errors (RMSD and ubRMSD) tend to remain rather stable. This indicates that the assimilation improves $\rho$ between model predictions and observations, but fails in reducing average errors in our experiment. This result is consistent with the findings of Rains et al. (2017).

To evaluate the effect of the assimilation on individual measurement points, Figure 7 shows the Taylor diagrams obtained
from the comparison between model predictions and *in situ* observations of soil moisture for two soil layer depths. In this figure, the circles indicate the open loop run performances and the triangles the assimilation run performances. Each colour is assigned to an individual observation point. In Figure 7 almost every individual observation point exhibits a $\rho$ improvement due to the assimilation and this for both soil layers. More precisely, all $\rho$ values increase for the first layer and all $\rho$ values except one increase for the second soil layer. The improvement is however rather different from one point measurement to another.
Moreover, Figure 7 indicates that, in general, the lower the open loop run $\rho$, the higher the improvement. This general feature is especially visible for the deeper soil layer.

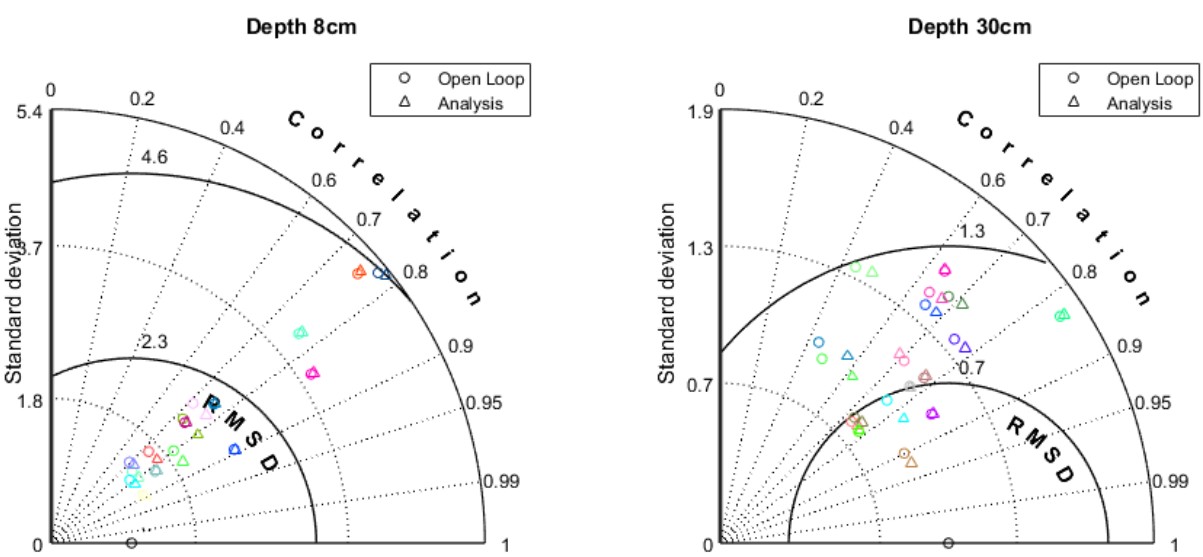

**Figure 7.** Taylor Diagrams (TD) drawn from the comparison between *in situ*-observed and SFX-simulated (background and analysis) soil moisture for the two different soil depth.

To analyse the spatial distribution of $\rho$ improvement as a result of the SMOS data assimilation, Figure 8 maps the $\rho$ changes between predictions and *in situ* measurements of soil moisture at all stations nd the temporal average of assimilation SM in-

crements (top panels), together with the average annual rainfall and Ep and the number of SMOS records assimilated over *in situ* soil moisture measurement sites (bottom panels). The two top panels show local $\rho$ improvements and SM increments in the 8 (top left panel) and 30 cm (top right panel) top soil layer respectively. The bottom panels in Figure 8 show the climate variability over the Murrumbidgee catchment using as a proxy the average annual rainfall and Ep (data provided by the

Australian Bureau of Meteorology), together with the number of SMOS records assimilated over *in situ* soil moisture measurement sites. In the bottom left panel of Figure 8, the same colour scale is used to indicate the rainfall amount (map) and the number of SMOS observations assimilated at each measurement station (colour of each points). The number of stations differs between left and right top panels as all stations do not measure soil moisture for all soil depths. The bottom panel indicates all measurement stations. Especially for the first layer, one can notice a gradient from East to West within the Murrumbidgee

basin (where observation sites are located). The two bottom panels in Figure 8 indicate that it is likely that the gradient in $\rho$ increase has its origin in climate variability but that it also depends on the the number of SMOS observations that are locally assimilated (panel on the bottom left hand side). In the Western semi-arid Murrumbidgee, soil moisture updates tend to have a longer-lasting effect on the performance because evapotranspiration is the main soil moisture controlling process and because the extraction of water from the soil due to evapotranspiration takes much longer than soil recharge due to rainfall. Moreover

one can notice in the top panels of Figure 8 that $\rho$ improvements are higher in areas where the temporal average of the absolute SM increments are higher and vice versa. This indicates that the $\rho$ is further improved when absolute SM increments tends to be higher.

   Overall, our experiment shows that the assimilation of SMOS data into the SFX model allows for a substantial improvement of the $\rho$ between model predictions and *in situ* observations of soil moisture with improvements similar to those obtained in a

very similar study by Rains et al. (2017) using the CLM land surface model.

   In Figure 9, we plot the averaged efficiency as a function of the open-loop soil moisture prediction percentiles for two soil depths to further investigate the effect of the assimilation on soil moisture prediction errors. To do so, we first compute, for each individual efficiency, the percentile of the synchronously obtained open loop soil moisture prediction and then compute the average efficiency for each percentile of the open loop soil moisture predictions. Figure 9 shows that the errors in soil moisture

prediction are mainly reduced by the assimilation for the higher quantiles of soil moisture while they tend to increase for the lower quantiles. For the upper layer, the assimilation is more efficient for predicted soil moisture values higher than the median. For the deeper layer, errors are reduced for quantiles higher than 80 %. This indicates that the assimilation is more efficient for high soil moisture states. A possible explanation for this is that the assimilation reduces errors when upper soil layers are closer to saturation, mainly during rainfall events when errors in ERA-Interim rainfall simulations are arguably affected by larger

errors. Although there is no absolute evidence that errors are larger for larger rainfall events for the Murray Darling basin, this is something that was often reported (for different areas of interest) in the literature as for example in the study by Xu et al. (2019).

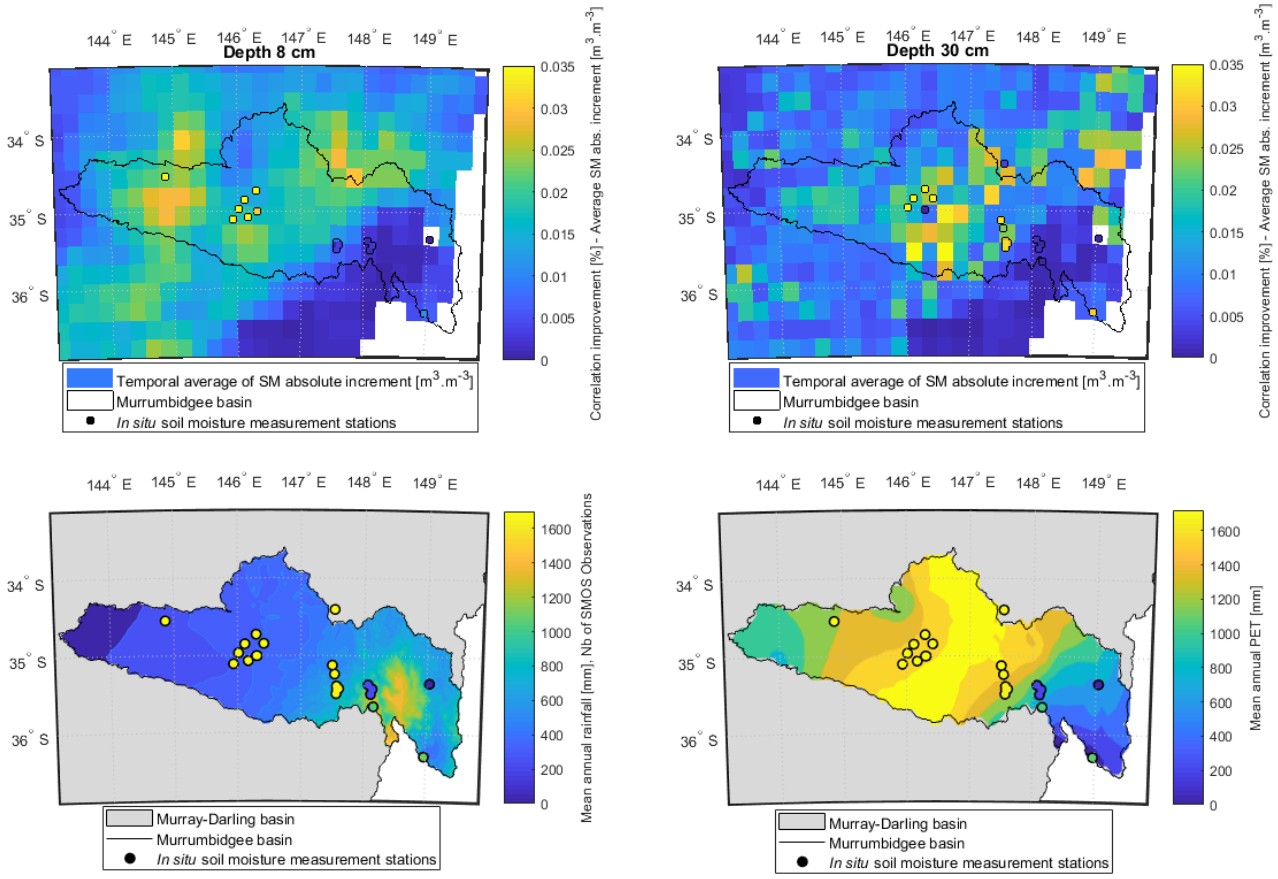

**Figure 8.** Correlation improvement in soil moisture prediction in relation to assimilation absolute increment, climate variability and number of assimilation events: Maps of the improvement in soil moisture $\rho$ and of the time average of absolute assimilation SM increments for the two root zone soil layers (upper panels) and maps of the number of assimilated SMOS Tb observation at each soil moisture measurement station (indicated via the dot colours), in relation with inter-annual average rainfall map (bottom left panel) and inter-annual average Ep map (bottom right panel).

## 3.4 Effect of the assimilation on predicted evapotranspiration

As evapotranspiration is also an important control in soil moisture dynamics, we propose to further evaluate the influence of the assimilation of SMOS observations on the monthly prediction of evapotranspiration. To do so, we compared the open loop and analysis simulations of monthly evapotranspiration with *in situ* observations derived from the flux towers (TERN OzFlux measurement network, http://www.ozflux.org.au/). Evapotranspiration observations are derived from monthly averaged flux tower measurements of latent heat flux. The comparison between observations and simulation results is carried out at the grid cells including the flux towers. The spatially averaged performance metrics yielded by this comparison are reported in Table 5, which revealed that the predictions of evapotranspiration are improved by the assimilation of SMOS observations as the $\rho$ with

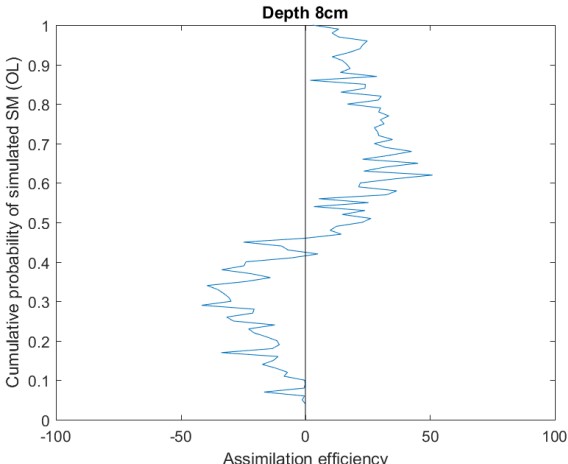
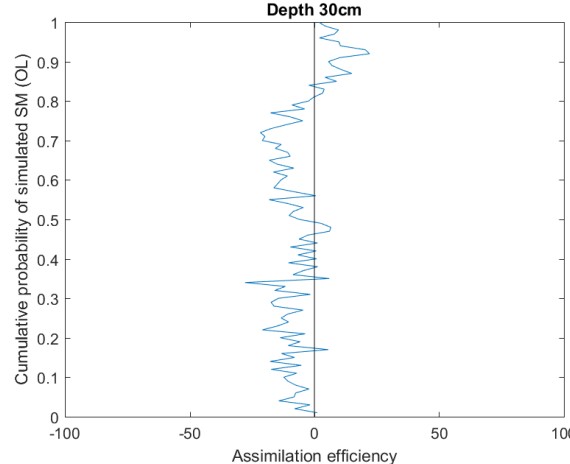

**Figure 9.** Assimilation efficiency as a function of simulated soil moisture quantiles: upper soil layer (left panel) and deeper soil layer (right panel).

**Table 5.** Time-space average values of background and analysis performances of evapotranspiration predictions (comparison with *in situ*-observed evapotranspiration).

|  | $\rho$ [-] | RMSD [mm.h$^{-1}$] | ubRMSD [mm.h$^{-1}$] |
|---|---|---|---|
| background | 0.46 | 0.057 | 0.034 |
| analysis | 0.48 | 0.056 | 0.034 |

*in situ* observations increased by 0.02 with a marginal reduction in RMSD. To assess the significance of the $\rho$ improvement as a result of the assimilation, we carried out Williams' significance tests (Williams, 1959). The corresponding p-values are all lower than 0.01 except for 1 station where the $\rho$ slightly decreases, indicating that the $\rho$ increase as a result of the assimilation of SMOS Tb can arguably be considered significant for most of the stations.

5    Figure 10 shows the Taylor diagram as well as the map of $\rho$ improvement for individual measurement stations and the temporal average of evapotranspiration increments. The effect of the assimilation on evapotranspiration is substantially positive for one station, limited for 3 of them and slightly negative for the last one. As for SM, one can notice in the right panel of Figure 10 that the $\rho$ is further improved when absolute evapotranspiration increments tends to be higher. Figure 11 shows the percentage improvement of simulated monthly evapotranspiration as a results of SMOS Tb for each individual flux tower

10   measurement together with averaged monthly rainfall (simulated by ERA-Interim).

In Figure 11, the assimilation lead from time to time either to an increase or a reduction of the error in simulated evapotranspiration. While the site having low annual precipitation (ca. 250 mm.yr$^{-1}$ at Calperum) exhibited a quasi systematic improvement, sites having medium annual precipitation (between ca. 480 to 560 mm.yr$^{-1}$ at Whroo, Riggs and Yanco) exhib-

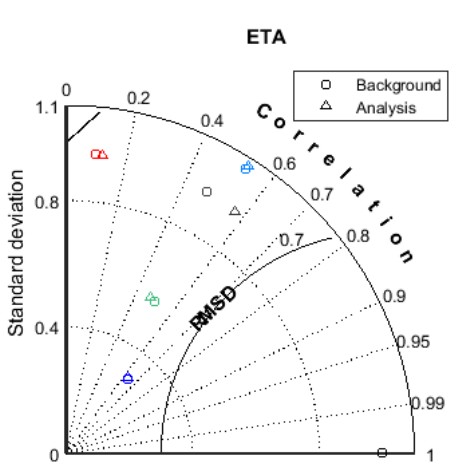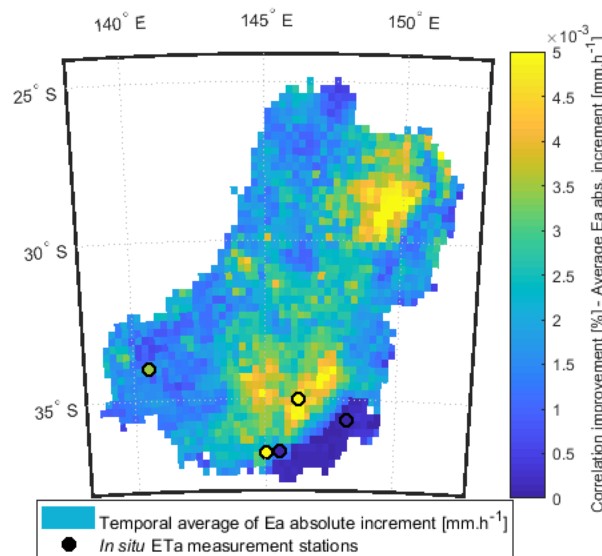

**Figure 10.** Effect of the assimilation on simulated evapotranspiration: Taylor diagram drawn from the comparison between *in situ*-observed and SFX-simulated evapotranspiration (Background and analysis, left panel) and map of the $\rho$ improvement and time average of absolute increments in evapotranspiration (right panel).

ited more contrasting results. The wettest site (ca. 720 mm.yr$^{-1}$ a t Tubarumba) showed very limited effect of the assimilation on the absolute error in simulated evapotranspiration. This result is in agreement with other studies (e.g., Detto et al., 2006; Vivoni et al., 2008; Mallick et al., 2018) that showed that water limitations in arid and semi-arid regions make evapotranspiration very sensitive to soil moisture variations, thereby explaining the fact that the assimilation of SMOS Tb is more efficient in
reducing errors of simulated evapotranspiration in water-limited regions of the MDB.

## 4   Conclusions

This study introduced and evaluated a large-scale SM modelling chain that is based on and takes advantage of the assimilation of SMOS Tb into a spatially distributed conceptual hydrological model coupled with a radiative transfer model. We assessed the performance of such a modelling chain and its associated data assimilation system and compared it with that of a quasi-
identical set up using the physically based CLM land surface model (Rains et al., 2017). We evaluated therefore whether a SM modelling chain, based on a conceptual hydrological model, is able to reach the same performance level as that of one based on a physically-based model, the main advantage of a conceptual model being its substantially lower computational demand. Eventually, we also evaluated how the assimilation of SMOS Tb can help in improving evapotranspiration predictions.

The following key conclusions can be drawn from our experiment:

1. A 6-year forward run of the SFX-based modelling chain reaches performance levels similar to those obtained with CLM both in terms of simulated Tb (comparison with SMOS data) and SM (comparison with *in situ* observation). The average

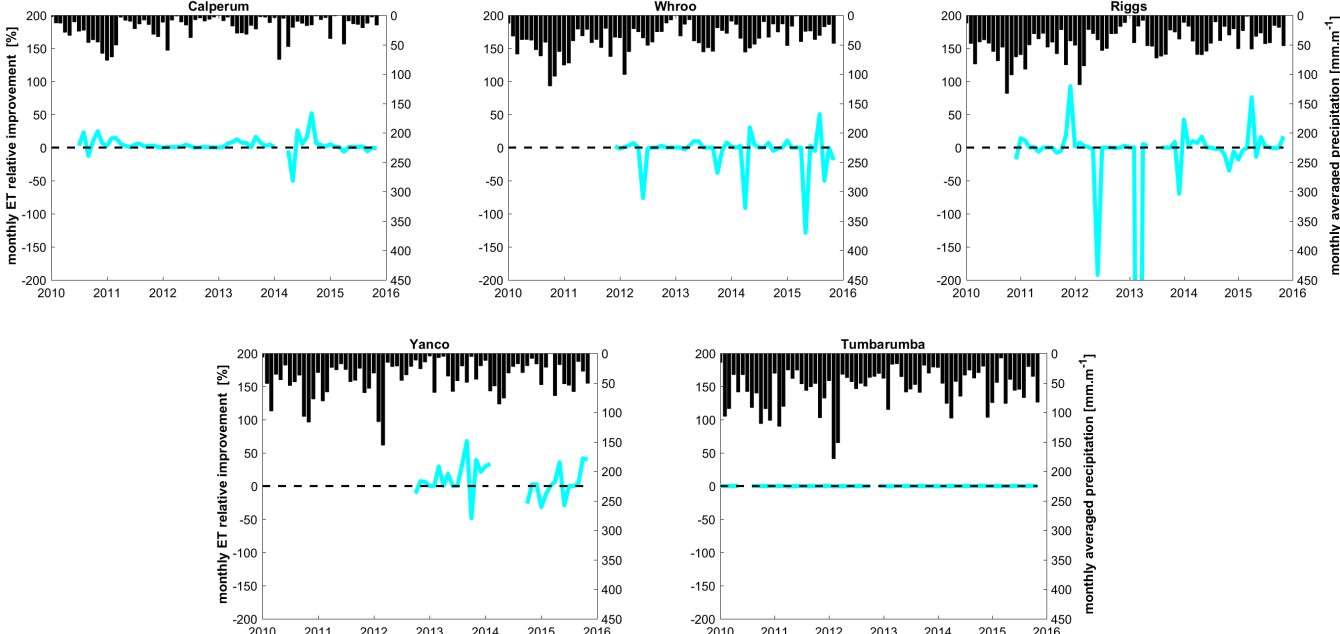

**Figure 11.** Effect of the assimilation on simulated evapotranspiration: Monthly rainfall and percentage improvement of simulated monthly evapotranspiration as a results of SMOS Tb assimilation for each individual flux tower measurement (flux tower locations sorted from west (top left panel) to east (bottom right panel).

$\rho$ values between simulated and SMOS observed Tb range between 0.62 and 0.72 for both models. The local $\rho$ values between simulated and *in situ* observed SM range between 0.3 and 0.8 for CLM and between 0.3 and 0.9 for SFX.

2. The assimilation of SMOS Tb observations into the SFX-based modelling chain increases the correlation between simulated and *in situ* observed SM by ca. 0.03.

3. The improvement in correlation between simulated and *in situ* observed SM as a result of the assimilation is slightly lower in our study than that obtained in Rains et al. (2017), but the correlation values are higher. As a result of the assimilation, the average correlations between simulated and *in situ* observed SM (top and deeper root zone soil layers) range between 0.65 and 0.68 for CLM and between 0.73 and 0.8 for SFX.

4. The assimilation of SMOS Tb observations reduces errors between simulated and *in situ* observed SM, especially for the highest SM values while it tends to increase them for lower SM values. For the upper layer, errors are reduced for SM values higher than the median. For the deeper layer, errors are reduced for quantiles higher than 80 %.

5. The assimilation of SMOS Tb observations increases correlation by 0.02 and marginally reduces errors between simulated and *in situ* observed evapotranspiration.

Overall, the study provides consistent empirical evidence that the SM modelling chain based on a conceptual hydrological model can reach and at times exceed the performance levels of a modelling chain based on a more physically based state of the art land surface model. While our assimilation experiment with SFX was carried out on a Personal Computer within a few hours, a High-Performance Computing cluster (using 2 nodes of 12 Cores) was necessary to run the Rains et al. (2017) experiment

over a few days. This shows the added value of a computationally efficient conceptual model, especially for applications where computational time is critical. Although the conceptual model needs to be calibrated, our experiment shows that this calibration can be carried out using only satellite data and has therefore the potential to be applicable to all areas where satellite data are reliable and informative.

*Code availability.* The code of the version of the Superflex model used for this study can be obtained upon request to the corresponding

author.

*Author contributions.* Renaud Hostache wrote the manuscript with support from all co-authors. All of the authors contributed to the design of the experiment and the analysis of the results. Renaud Hostache implemented the codes necessary for the experiments. Dominik Rains provided the datasets and contributed to the comparison between the SFX- and the CLM-based experiments.

*Competing interests.* The authors declare that they have no conflict of interest.

*Acknowledgements.* The research reported herein was funded by the National Research fund of Luxembourg and BelSPO through the Hydras+ (SR/8888433, SR/00/302) and the CASCADE (C17/SR/11682050) projects.

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
