# Peer review of "Assimilation of SMOS brightness temperature into a large-scale distributed conceptual hydrological model to improve soil moisture predictions: the Murray-Darling basin in Australia as a test case."

_Hydrology and Earth System Sciences, 2019_

## Referee Comment (RC1) · Anonymous Referee #1 · 1 Sep 2019

**OVERVIEW**

The study investigates the use of a distributed conceptual hydrological model for simulating soil moisture (and evaporation) over large areas. Specifically, the model has been applied over the Murray-Darling basin in Australia and calibrated by using brightness temperature observations from SMOS. For that, the hydrological model (SUPERFLEX) is coupled with a radiative transfer model (CMEM). The model validation with in situ soil moisture (and evaporation) data has been carried out and compared with the

CLM-CMEM simulation performed in Rains et al. (2017).

**GENERAL COMMENTS**

The paper is fairly well written and clear. The topic is of interest for the readership of Hydrology and Earth System Sciences journal. The use of satellite measurements for calibrating hydrological modelling is an important topic and the development of new approaches for addressing the task is surely of interest. Therefore, I believe the paper might deserve to be published but, in my opinion, after the clarification of some important points.

I listed here the main comments also including their relevance:

1) MAJOR: The main assumption of the paper is that a distributed and conceptual hydrological model is more flexible, easier to use, less complex and faster than a land surface model. Therefore, if with a calibrated hydrological model we obtain similar performance as compared with a land surface model, we can build better modelling approaches. However, the assumptions above are not tested. Several questions come to my mind.

a. Is the hydrological model SUPERFLEX less complex than CLM? The structure of the two models should be shown and compared. I have the feeling that the model complexity is nearly the same.

b. Is SUPERFLEX faster than CLM? Some information on the running time for the two modelling approaches should be given.

c. Why do we need a faster and less complex model? Which applications are addressed? For climate applications, we don't need faster simulations, right?

I believe these questions should be addressed before the publication.

2) MAJOR: Is CLM calibrated on SMOS observations? As I believe it is not the case

(from Rains et al., 2017), it is unexpected that CLM and SUPERFLEX perform similarly for the reproduction of SMOS brightness temperature (SUPERFLEX is calibrated on SMOS). Do the authors have an explanation? Similarly, results against in situ soil moisture observations are similar suggesting that CLM is performing good also without calibration. SUPERFLEX is calibrated with SMOS brightness temperature that in Australia is well correlated with in situ soil moisture (from previous studies), therefore I expect it works good against in situ soil moisture. On this basis, I believe CLM should be considered more reliable than SUPERFLEX. Can the authors comment on that?

3) MODERATE: The differences between the open loop and the analysis are very small. Are they significant? Some tests to assess the significance of the obtained results should be performed.

4) MODERATE: Does the SUPERFLEX model include lateral flow? If not (as I believe), it should be clarified.

I listed in the specific comments a number of corrections and changes that are needed.

**SPECIFIC COMMENTS (P: page, L: line or lines)**

P1, Abstract: In the abstract, I have found too many details on the methodology and just few lines for the results. E.g., the results for simulating actual evapotranspiration are not mentioned. Please revise the abstract.

P2, L5-10: In the introduction, the prediction of flood is mentioned but the modelling approach here developed is tested only in terms of soil moisture and evapotranspiration. Indeed, I expected to see also results in terms of river discharge simulations by reading the title (hydrological model). Anyhow, less emphasis should be given to flood forecasting in the introduction.

P2, L32: Acronyms should be defined, and references to modelling approaches should

be given. Throughout the text, the acronyms should be defined.

P3, L5: It should be "2009" instead of "2019".

P3, L8: Please rename the "land surface modelling" for retrieval of soil moisture from SMOS brightness temperature. It makes confusion with land surface model. I would rename in "radiative transfer modelling".

P4, L1: "tailoring the structure" of? Please clarify.

P5, Figure 1: Map of Australia should be smaller, and that of Murray-Darling bigger.

P5, L1: I understand the use of ERA-Interim for performing the simulation as in Rains et al. (2017); however, it would be highly interesting to test ERA5, the new ECMWF reanalysis.

P7, L5: It shouldn't be "surface" runoff, but total runoff, right?

P7, L9: not bold for "URl".

P12, L14-15: Is the gradient of performance of SUPERFLEX similar to CLM? Please comment on that.

P13, L1: Use "target" instead of "0" to avoid misunderstanding with the axis origin.

P14, L5-7: Why is there a strong underestimation of standard deviation? Do the authors have some explanations?

P16, L14: Why $25K^2$? Please add a reference or an explanation.

P16, L14: "average performance metric" as compared with? Please clarify.

P19, L1-: This part should be moved to the method section.

P20, L13-14: Are we sure that ERA-Interim rainfall has larger errors for larger rainfall events?

P21, L9: This part should be moved to the method section.

P23, L6-17: There is no need to summarize the study in the conclusions; I suggest shortening this part.

**RECOMMENDATION**

On this basis, I found the topic of the paper relevant, and I suggest a major revision before the paper can be published on Hydrology and Earth System Sciences.

---

## Referee Comment (RC2) · Anonymous Referee #2 · 8 Oct 2019

The main objectives of this study is to compare a large-scale conceptual hydro-meteorological model (SUPERFLEX) and a physically-based land surface models (CLM) in their ability to simulate SMOS-like brightness temperature (Tb) and soil moisture, and (ii) to evaluate the improvement in model predictions when assimilating SMOS Tb observations. It is well written and the abstract reflects the objectives and results well. Results are supported by appropriate figures and tables, references (some could be updated). This study is very interesting and promising. However, the paper does not do it justice. I feel like it was written quickly and that the authors skimmed over some key explanations. A more in-depth explanation of the methodology and analysis of the results are necessary. They are some caveats and I am particularly concerned

about the title that does not reflect the real purpose of this study (comparison the ability of SUPERFLEX and CLM to simulate Tb and soil moisture). To me this kind of comparison is a bit unfair from the beginning as you do not necessarily want a large-scale distributed conceptual hydrological model and a large scale physically based land surface model (CLM) for the same purpose (?) Not to mention the fact that SUPERFLEX is calibrated, what about CLM? If authors wish to pursue in this way, then I am missing a proper description of the CLM set up (not only referring to a previous study and mentioning a 'quasi-identical' set up). My recommendation is major review, please find below an attempt to help.

General major comments (additionally to what is mentioned above) -From the abstract I see scores and headlines but I have no clue where the study takes place, please introduce south eastern Australia from the beginning (maybe from the title, it has to change anyway to reflect the content of the work).

-Also from the abstract it is surprising that ERA-Interim is still used rather than ERA5. The recent literature (2018, 2019) is already full of studies demonstrating the added value of ERA5 with respect to ERA-Interim. I assume that in the previous CLM study (Rains et al., 2017) ERA-Interim was used and that is the rational for keeping it. This should be stated somewhere and ERA5 mentioned, if not tested as I believe it will prove useful.

- work must be done on statistical scores to provide an indication of how significant they are, I suggest to add at least p-values to assess the significance of each datasets and a 95% confidence interval (based on boot strapping?) to assess either or not differences from the 2 configurations are significant (that can possibly hamper your conclusions?).

-Figures and tables should be self explanatory (?) please expand captions, add units when necessary (Kelvin...), label each panels for sake of clarity and refer to the labelling in the captions (some figure are hardly visible).

Other comments - scores from the abstract should more detailed, are you talking about

surface soil moisture? Root zone soil moisture?

- P.2, L.13, '[...] as uncertain forcing [...]' OK so justify the use of ERA-Interim over ERA5

- P.2, L.27, surface soil moisture (SSM)

- P.3, L.5, November 2019 ? Do you mean 2009 ?

- P.14, L.15, "[...] is impacting soil moisture variations more significantly [...]" what is the meaning of "significantly"?

- Figure 2 must be improved, ground based measurement stations are barely visible, also is the main river represented the only one in Australia (this is not a paper quality figure).

- section 2.2.1, if this work has been published elsewhere, maybe it can be put in an annexe / supplementary ?

- P.7, L.2, "[...] 0.25° matching the one used in ERA-Interim dataset." misleading at you put is at 0.25° while its native spatial resolution is closer to 80km

- I am missing somewhere a clear description of the 2 models set up

- Section 2.2.3, please discuss further the possible impact on the 2 models comparison.

- Figure 3, units, significance, labels

- Table 1, as it stands it is not very useful, expand the caption so readers may know what is it about, statistics between what and what? What are the units? Cal stands for calibration, Val stands for validation...(general major comment), use same number of digit..

- Figure 4, is left panel useful? Significance of the differences in figure 5?

- P.15, L.6, "[...] time series [...]" a figure would prove useful

**HESSD**

- Figure 6, same min/max for axis of left and right panels

- P.17, L.9-12, please discuss the use of SMOS anomalies as it could be an explanation (?)

- Figure 8, I do not understand bottom left panel, rainfall and #obs? Also why the number of stations differs from a panel to another?

- Figure 9 is interesting!

- Figure 10, not clear enough that 5 pairs of data are represented on the Taylor Diagram, pleas improve the quality.

- Title presents 1 objective, the abstract 2 and the beginning of the conclusion 3, please be consistent.

- I am personally not a big fan of bullet points in a conclusion but I may be a personal statement.

Thanks!

---

## Referee Comment (RC3) · Anonymous Referee #3 · 24 Nov 2019

I fully agree with the author to try out the usability of a conceptual hydrological model (when compared to detailed LSMs) assimilating TB, in terms of capturing hydrological processes at large scale. Such concept of spatially calibration (or DA to enable optimal estimate) is very innovative to overcome certain calibration problems of distributed hydrological model with only one stream flow gauge. I do have several comments for the authors to consider: 1. The manuscript focused on SFX model simulations and their coupling with CMEM as well as the whole setup in the DA framework, as well as the results comparison with the previous study using CLM. Although it is understandable that the author try to maintain the quasi similar setup to enable the consistent comparison, the current presentation of such 'similarity' is not clear. Lots of details

need to dig out by readers via reading the paper of Rains. This reviewer is wondering if the author can make better presentations on this perspective, either via a diagram, via summary tables, etc. 2. It is very good to see the forward simulation comparisons between SFX-based and CLM-based models. On the other hand, the advantage of assimilating SMOS TB on soil moisture and ET is not presented in a satisfied manner. The manuscript only shows the performance for the in-situ sites, but not the spatial pattern changes of soil moisture and ET (before and after assimilating Tb). This point should be addressed to show straightforwardly the advantage of incorporating spatial information for distributed conceptual hydrological model simulations. 3. It is not clear if the author only run the model at the timestep of satellite observation intervals or also run the model in between observations? E.g., if the model only start to run with the satellite observation when they are available, then the author missed a lots of details in between satellite observations It is assumed with the forcing the model can get more frequent simulation results as outputs to capture hydrological processes? 4. Last but not least, it is strongly recommended to have a native English editor to go through the manuscript. Some specific comments please find attached PDF.

Please also note the supplement to this comment:
https://www.hydrol-earth-syst-sci-discuss.net/hess-2019-414/hess-2019-414-RC3-supplement.pdf

**Supplement:**

[revised manuscript text omitted]

---

## Author Comment (AC1) · 24 Jan 2020

We would like first to thank Referee 1 for the careful reading of the paper and the relevant remarks and comments. In the remainder, Referee 1's remarks are written in normal font while **our answers are written in bold**

GENERAL COMMENTS The paper is fairly well written and clear. The topic is of interest for the readership of Hydrology and Earth System Sciences journal. The use of satellite measurements for calibrating hydrological modelling is an important topic and the development of new approaches for addressing the task is surely of interest. Therefore, I believe the paper might deserve to be published but, in my opinion, after

the clarification of some important points.

**We thank Referee 1 for this assessment.**

I listed here the main comments also including their relevance: 1) MAJOR: The main assumption of the paper is that a distributed and conceptual hydrological model is more flexible, easier to use, less complex and faster than a land surface model. Therefore, if with a calibrated hydrological model we obtain similar performance as compared with a land surface model, we can build better modelling approaches. However, the assumptions above are not tested. Several questions come to my mind.

a. Is the hydrological model SUPERFLEX less complex than CLM? The structure of the two models should be shown and compared. I have the feeling that the model complexity is nearly the same.

**A better description of CLM will be provided in the updated version of the paper. We argue that the SUPERFLEX model is less complex than CLM for the following reasons: 1. Whereas CLM solves dedicated equations for energy balance taking into account vegetation status, SUPERFLEX lumps energy balance contribution to water balance via a simpler potential evapotranspiration formula. 2. Superflex has a limited number of parameters compared to CLM 3. In this study, SUPER-FLEX is structured with two soil layers (respectively 0-7cm and 7-21cm) whereas CLM considers five layers for the same soil depth range. Moreover, Superflex has a limited number of parameters compared to CLM. The Set up of CLM therefore requires much more input data (e.g. soil types and land use IIRC). These are the principal differences (but not the only ones) between the CLM and the SUPERFLEX set up that, in our opinion, enable us to argue that SUPERFLEX is less complex than CLM. This will be better explained in the revised version of our manuscript.**

b. Is SUPERFLEX faster than CLM? Some information on the running time for the two modelling approaches should be given. **Yes, it is much faster. More information on**

**this will be added in the revised version of the paper.**

c. Why do we need a faster and less complex model? Which applications are addressed? For climate applications, we don't need faster simulations, right? I believe these questions should be addressed before the publication.

**We argue that faster models are a pre-requisite for carrying simulations at large scale without implying a high computational demand. Moreover, we argue that faster models are required for near real-time forecasting applications and for long-term simulations. This will be highlighted in the new version of the article.**

2) MAJOR: Is CLM calibrated on SMOS observations? As I believe it is not the case (from Rains et al., 2017), it is unexpected that CLM and SUPERFLEX perform similarly for the reproduction of SMOS brightness temperature (SUPERFLEX is calibrated on SMOS). Do the authors have an explanation? Similarly, results against in situ soil moisture observations are similar suggesting that CLM is performing good also without calibration. SUPERFLEX is calibrated with SMOS brightness temperature that in Australia is well correlated with in situ soil moisture (from previous studies), therefore I expect it works good against in situ soil moisture. On this basis, I believe CLM should be considered more reliable than SUPERFLEX. Can the authors comment on that?

**SUPERFLEX is indeed calibrated using SMOS observations. Referee 1 is right; CLM is not calibrated using SMOS data. Indeed, while a conceptual model such as SUPEFLEX requires a calibration effort because its parameter values cannot be set a priori, CLM is not supposed to be calibrated as it is physically based and its parameters are derived from various input data describing the characteristics of the catchment. Moreover, one can argue that, because of a large number of parameters, calibrating CLM using SMOS data would not be an easy task especially due to the computational demand over a large basin such as the Murray Darling. The calibration of many parameters would lead to a widely reported equifinality issue. We fully agree that CLM performs satisfyingly even without**

any calibration. However, we think Referee 1 slightly misunderstood the objective of our study: we want to evaluate if a simplistic conceptual model when calibrated with a freely and globally available data product such as SMOS Tb can reach the performance level of a physically based model. We believe that the conclusions of our paper show that this is actually true. However, we do not agree with the statement that CLM should be considered more reliable than SUPERFLEX as the performance level are rather similar for both models during the calibration and the validation period. The advantage of CLM is that it does not require any calibration, while the strength of SUPERFLEX is that it can reach the same level of performance when calibrated with a freely and globally available SMOS data product. In addition, it can reach this performance level with a comparatively lower computational demand. We will clarify this as requested by Referee 1.

3) MODERATE: The differences between the open loop and the analysis are very small. Are they significant? Some tests to assess the significance of the obtained results should be performed.

**To answer this question, we will carry out and present significance tests before resubmitting the revised version of the manuscript.**

4) MODERATE: Does the SUPERFLEX model include lateral flow? If not (as I believe), it should be clarified.

**This is right; In the simplified version used in this experiments SUPERFLEX does not explicitly simulate lateral flows within the root zone soil layers. This will be clarified in the revised version of the paper.**

I listed in the specific comments a number of corrections and changes that are needed. SPECIFIC COMMENTS (P: page, L: line or lines) P1, Abstract: In the abstract, I have found too many details on the methodology and just few lines for the results. E.g., the results for simulating actual evapotranspiration are not mentioned. Please revise the

abstract.

**We thank Referee 1 for this relevant remark. We will edit the abstract accordingly**

P2, L5-10: In the introduction, the prediction of flood is mentioned but the modelling approach here developed is tested only in terms of soil moisture and evapotranspiration. Indeed, I expected to see also results in terms of river discharge simulations by reading the title (hydrological model). Anyhow, less emphasis should be given to flood forecasting in the introduction.

**We thank Referee 1 for this relevant remark. We will edit the introduction accordingly**

P2, L32: Acronyms should be defined, and references to modelling approaches should be given. Throughout the text, the acronyms should be defined.

**We will define the acronyms.**

P3, L5: It should be "2009" instead of "2019".

**Thanks for pinpointing us to this mistake**

P3, L8: Please rename the "land surface modelling" for retrieval of soil moisture from SMOS brightness temperature. It makes confusion with land surface model. I would rename in "radiative transfer modelling".

**This will be done**

P4, L1: "tailoring the structure" of? Please clarify.

**We will edit the text so that it becomes clearer that the structure of the model (i.e. reservoirs. . .) can be tailored.**

P5, Figure 1: Map of Australia should be smaller, and that of Murray-Darling bigger.

**This will be done as requested by Referee 1**

P5, L1: I understand the use of ERA-Interim for performing the simulation as in Rains et al. (2017); however, it would be highly interesting to test ERA5, the new ECMWF reanalysis.

**We agree that this would be interesting in general. However, this goes beyond the scope of the paper as the new results would not allow for a meaningful comparison with the study by Rains et al (2017) which is one of the main objectives of this study. We are happy to consider this remark for further studies.**

P7, L5: It shouldn't be "surface" runoff, but total runoff, right?

**This will be clarified: In the sentence, "surface runoff" was actually related to "routing function" (that simulates surface runoff)**

P7, L9: not bold for "URI".

**Thanks, this will be corrected.**

P12, L14-15: Is the gradient of performance of SUPERFLEX similar to CLM? Please comment on that.

**The gradient of performance of SUPERFLEX that is visible in Fig.3 is not observed in CLM's run. As argued in the paper: Considering that in our set up, the representation of the evapotranspiration is rather simplistic as it is based on the Hamon formula, this could explain the comparatively poor performance of the model in the western part of the basin.**

P13, L1: Use "target" instead of "0" to avoid misunderstanding with the axis origin.

**This will be modified**

P14, L5-7: Why is there a strong underestimation of standard deviation? Do the authors have some explanations?

**Our interpretation is that the two models are unable to reproduce the variance**

**of SMOS observations mainly due to some limitations of the radiative transfer model. Indeed, even with completely dry or wet soils, the simulated Tb do not reach the extreme values observed by SMOS.**

P16, L14: Why 25K2? Please add a reference or an explanation.

**Here, we used the same value as in Rains et al. (2017) to keep the experiments comparable.**

P16, L14: "average performance metric" as compared with? Please clarify.

**The reference used for this comparison are the in situ soil moisture measurements. This will be further clarified in the revised version of the paper.**

P19, L1-: This part should be moved to the method section.

**This will be modified in the revised version of the paper**

P20, L13-14: Are we sure that ERA-Interim rainfall has larger errors for larger rainfall events?

**We cannot be sure that errors are larger for larger rainfall for the Murray Darling basin, but this is something that was often reported in the literature as for example in the study by Xu et al (2019). Xiaoyong Xu, Steven K. Frey, Alaba Boluwade, Andre R. Erler, Omar Khader, David R. Lapen, Edward Sudicky, Evaluation of variability among different precipitation products in the Northern Great Plains, Journal of Hydrology: Regional Studies, Volume 24, 2019.**

P21, L9: This part should be moved to the method section.

**This will be modified in the revised version of the paper**

P23, L6-17: There is no need to summarize the study in the conclusions; I suggest shortening this part. **The conclusion part will be shortened accordingly**

414, 2019.

---

## Author Comment (AC2) · 24 Jan 2020

We would like first to thank Referee 2 for the careful reading of the paper and the relevant remarks and comments. In the remainder, Referee 2's remarks are written in normal font while **our answers are written in bold**.

The main objectives of this study is to compare a large-scale conceptual hydrometeorological model (SUPERFLEX) and a physically-based land surface models (CLM) in their ability to simulate SMOS-like brightness temperature (Tb) and soil moisture, and (ii) to evaluate the improvement in model predictions when assimilating SMOS Tb observations. It is well written and the abstract reflects the objectives and results well.

Results are supported by appropriate figures and tables, references (some could be updated). This study is very interesting and promising. However, the paper does not do it justice. I feel like it was written quickly and that the authors skimmed over some key explanations. A more in-depth explanation of the methodology and analysis of the results are necessary. They are some caveats and I am particularly concerned about the title that does not reflect the real purpose of this study (comparison the ability of SUPERFLEX and CLM to simulate Tb and soil moisture). To me this kind of comparison is a bit unfair from the beginning as you do not necessarily want a large-scale distributed conceptual hydrological model and a large scale physically based land surface model (CLM) for the same purpose (?) Not to mention the fact that SUPERFLEX is calibrated, what about CLM? If authors wish to pursue in this way, then I am missing a proper description of the CLM set up (not only referring to a previous study and mentioning a 'quasi-identical' set up). My recommendation is major review, please find below an attempt to help.

**We thank Referee 2 for this analysis. A more detailed description of CLM will be provided in the updated version of the paper. SUPERFLEX is calibrated whereas CLM is not. Indeed, while a conceptual model such as SUPEFLEX requires a substantial calibration effort because its parameter values cannot be set a priori, CLM is not supposed to be calibrated as it is physically based and its parameters can be determined based on the basin's known characteristics. Moreover, one can argue that calibrating CLM using SMOS data would not be an easy task, especially because of a high number of parameters and a significant computational effort needed to achieve a robust calibration for a large basin such as the Murray Darling. We will clarify this in the article.**

General major comments (additionally to what is mentioned above) -From the abstract I see scores and headlines but I have no clue where the study takes place, please introduce south eastern Australia from the beginning (maybe from the title, it has to change anyway to reflect the content of the work).

**We thank Referee 2 for this relevant remark. We will edit the abstract and the title accordingly**

-Also from the abstract it is surprising that ERA-Interim is still used rather than ERA5. The recent literature (2018, 2019) is already full of studies demonstrating the added value of ERA5 with respect to ERA-Interim. I assume that in the previous CLM study (Rains et al., 2017) ERA-Interim was used and that is the rational for keeping it. This should be stated somewhere and ERA5 mentioned, if not tested as I believe it will prove useful.

**We agree that ERA 5 is a better dataset than ERA interim and is of great interest. However, as acknowledged by Referee 2, using ERA5 for this study would render any comparison with the study by Rains et al. (2017) rather meaningless. We consider this comparison to be one of the main objectives of this study. We will better explain why we used ERA-interim and introduce Era5 in the paper.**

- work must be done on statistical scores to provide an indication of how significant they are, I suggest to add at least p-values to assess the significance of each datasets and a 95

**We will carry out and present significance tests for the experiment based on SUPERFLEX before resubmitting the revised version of the manuscript.**

-Figures and tables should be self explanatory (?) please expand captions, add units when necessary (Kelvin: : :), label each panels for sake of clarity and refer to the labelling in the captions (some figure are hardly visible).

**We will clarify and improve some of the figures and captions accordingly.**

Other comments - scores from the abstract should more detailed, are you talking about surface soil moisture? Root zone soil moisture?

**The scores given in the abstract have been computed based on observed soil moisture for both the surface layer and the 30cm root zone soil layer. This will**

**be clarified in the revised version of the article.**

- P.2, L.13, '[...] as uncertain forcing [...]' OK so justify the use of ERA-Interim over ERA5

**We agree that ERA 5 is a better dataset than ERA interim and is of great interest. However, as acknowledged by Referee 2, using ERA5 for this study would render any comparison with the study by Rains et al. (2017 rather meaningless. We will better explain why we used Era-interim and introduce Era5 in the paper.**

- P.2, L.27, surface soil moisture (SSM)

**We will add Âń surface Âż before soil moisture accordingly.**

- P.3, L.5, November 2019 ? Do you mean 2009 ?

**This is right . Thanks for highlighting this mistake**

- P.14, L.15, "[: : :] is impacting soil moisture variations more significantly [...]" what is the meaning of "significantly"?

**Here we meant to say that evapotranspiration has a higher impact on soil moisture variations.**

- Figure 2 must be improved, ground based measurement stations are barely visible, also is the main river represented the only one in Australia (this is not a paper quality figure).

**Our assumption is that Referee 2 is referring to figure 1 rather than figure 2. The right panel of this figure will be enlarged to render the measurement stations more visible. We will improve this figure in general.**

- section 2.2.1, if this work has been published elsewhere, maybe it can be put in an annexe / supplementary ?

**To our knowledge this work has not been published elsewhere**

- P.7, L.2, "[ : : :] $0.25_m atching the one used in ERA - Interim dataset." misleading at you put is at $0.25_w$ hile its native spatial resolution is closer to 80km

**We thank Referee 2 for highlighting this. We will clarify this point in the revised version of the paper.**

- I am missing somewhere a clear description of the 2 models set up

**A description of CLM will be provided in the updated version of the paper.**

- Section 2.2.3, please discuss further the possible impact on the 2 models comparison.

**We will clarify in the paper that SUPERFLEX is calibrated and CLM not. We will also emphasise the fact that the calibration of Superflex using SMOS data gives it a higher chance to reach better performance levels. Moreover, we will argue that calibrating CLM using SMOS data would not be an easy task, especially due to the required computational demand for calibrating a very high number of parameters over a large basin such as the Murray Darling.**

- Figure 3, units, significance, labels

**This will be done**

- Table 1, as it stands it is not very useful, expand the caption so readers may know what is it about, statistics between what and what? What are the units? Cal stands for calibration, Val stands for validation...(general major comment), use same number of digit..

**We will improve the caption of this table.**

- Figure 4, is left panel useful? Significance of the differences in figure 5?

**Yes as the left panel proposes average values that are more integrative, we believe it is useful.**

- P.15, L.6, "[: : :] time series [...]" a figure would prove useful

**A figure with a time series would only show results for one station. We think that showing a more general analysis using Taylor diagrams is more relevant.**

- Figure 6, same min/max for axis of left and right panels

**Since the considered soil depths differ, the performance value ranges in different intervals. We do not see the benefit of "same min/max for axis of left and right panels" as this would render the figure less readable (i.e. the points will be closer to each other).**

- P.17, L.9-12, please discuss the use of SMOS anomalies as it could be an explanation (?)

**The fact that we assimilate SMOS anomalies helps reducing bias between simulations and observations during the assimilation. However, at that moment, we do not see why the assimilation of anomalies would explain the fact that ubRMSE and RMSE are not reduced.**

- Figure 8, I do not understand bottom left panel, rainfall and obs? Also why the number of stations differs from a panel to another?

**In the bottom left panel, the same colour scale is used to indicate rainfall amounts (map) and number of SMOS observations assimilated at each measurement station (colour of each points). This will be clarified in the new version of the paper. The number of stations differs from one panel to another as all stations do not measure soil moisture for all soil depths. The bottom panel indicates all measurement stations. The top panels show stations measuring SM in the 8 and 30 cm top soil layer. This will be clarified as well.**

- Figure 9 is interesting!

**We thank Referee 2 for this positive remark.**

- Figure 10, not clear enough that 5 pairs of data are represented on the Taylor Diagram, pleas improve the quality.

**We will improve this figure**

- Title presents 1 objective, the abstract 2 and the beginning of the conclusion 3, please be consistent.

**We will improve this**

- I am personally not a big fan of bullet points in a conclusion but I may be a personal statement.

**We thank Referee 2 for this remark.**

---

## Author Comment (AC3) · 24 Jan 2020

We would like first to thank Referee 3 for the careful reading of the paper and the relevant remarks and comments. In the remainder, Referee 3's remarks are written in normal font while **our answers are written in bold**.

I fully agree with the author to try out the usability of a conceptual hydrological model (when compared to detailed LSMs) assimilating TB, in terms of capturing hydrological processes at large scale. Such concept of spatially calibration (or DA to enable optimal estimate) is very innovative to overcome certain calibration problems of distributed hydrological model with only one stream flow gauge.

**We thank Referee 3 for this assessment.**

I do have several comments for the authors to consider: 1. The manuscript focused on SFX model simulations and their coupling with CMEM as well as the whole setup in the DA framework, as well as the results comparison with the previous study using CLM. Although it is understandable that the author try to maintain the quasi similar setup to enable the consistent comparison, the current presentation of such 'similarity' is not clear. Lots of details need to dig out by readers via reading the paper of Rains. This reviewer is wondering if the author can make better presentations on this perspective, either via a diagram, via summary tables, etc.

**We thank Referee 3 for highlighting this. We will put efforts on further explaining and clarifying the similarities and the differences between the two experiments. We will in particular add a better description of Rains et al (2017)'s experiment and CLM model.**

2. It is very good to see the forward simulation comparisons between SFX-based and CLM-based models. On the other hand, the advantage of assimilating SMOS TB on soil moisture and ET is not presented in a satisfied manner. The manuscript only shows the performance for the in-situ sites, but not the spatial pattern changes of soil moisture and ET (before and after assimilating Tb). This point should be addressed to show straightforwardly the advantage of incorporating spatial information for distributed conceptual hydrological model simulations.

**According to Referee 3's suggestion, we propose to add figures with assimilation increments in the revised version of the paper to further investigate the effect of the Tb assimilation on spatial patterns of SM and ET.**

3. It is not clear if the author only run the model at the timestep of satellite observation intervals or also run the model in between observations? E.g., if the model only start to run with the satellite observation when they are available, then the author missed a lots of details in between satellite observations It is assumed with the forcing the model

can get more frequent simulation results as outputs to capture hydrological processes?

**We apologize that this was not clear enough. The ensemble of models run at an hourly time step as well in-between SMOS observations. We will clarify this point in the revised version of the article**

4. Last but not least, it is strongly recommended to have a native English editor to go through the manuscript. Some specific comments please find attached PDF.

**According to Referee 3's suggestion, a native English speaker will carefully read and edit our paper before its resubmission. The Specific comments in the supplementary review material provided by Referee 3 will be carefully addressed in the new version of the paper.**

---

## Author Response (AR1)

This document contains our answers to the referee's and editor's comments. We would like first to thank Referees #1, 2 and 3 and the editor for the careful reading of the paper and their relevant remarks and comments. In the remainder, Referee's remarks are written in black while our answers are written in blue. Moreover, cited text from the revised version of the paper is written in red.

Enclosed are as well two versions of our revised paper: one without track change and one with track change

**Referee #1**

GENERAL COMMENTS
The paper is fairly well written and clear. The topic is of interest for the readership of Hydrology and Earth System Sciences journal. The use of satellite measurements for calibrating hydrological modelling is an important topic and the development of new approaches for addressing the task is surely of interest. Therefore, I believe the paper might deserve to be published but, in my opinion, after the clarification of some important points.

We thank Referee #1 for his assessment.

I listed here the main comments also including their relevance:
1) MAJOR: The main assumption of the paper is that a distributed and conceptual hydrological model is more flexible, easier to use, less complex and faster than a land surface model. Therefore, if with a calibrated hydrological model we obtain similar performance as compared with a land surface model, we can build better modelling approaches. However, the assumptions above are not tested. Several questions come to my mind.

a. Is the hydrological model SUPERFLEX less complex than CLM? The structure of the two models should be shown and compared. I have the feeling that the model complexity is nearly the same.

To clarify these points we add a paragraph highlighting some key differences in the revised version of the paper.

From the SFX model description, it appears that this conceptual model is much simpler than CLM in particular for the following reasons:
- Whereas CLM estimates latent heat fluxes between soil and atmosphere based on aerodynamic diffusion equation and Monin-Obukhov similarity theory Oleson et al. 2013), taking into account many information such as soil and vegetation type, surface roughness, atmospheric stability, the vegetation coverage, SUPERFLEX lumps energy balance contribution to water balance via a simpler potential evapotranspiration formula, namely the Hamon formula.
- In this study, SFX is structured with two soil layers (respectively 0-7cm and 7-21cm) whereas CLM considers five layers for the same soil depth range.
- The set up of CLM therefore requires much more input data (e.g., soil types and land use), and Superflex has a limited number of parameters (6 parameters per cell) compared to CLM (potentially tens of parameters per cell, Hou et al., 2012).It is worth noting here that this list is not exhaustive and that many other processes are further simplified in SFX compared to CLM. A detailed presentation of CLM is available in Oleson et al. (2013).

These are the principal differences (but not the only ones) between the CLM and the SUPERFLEX set up that, in our opinion, enable us to argue that SUPERFLEX is less complex than CLM. As shown by the added text above, this is now better explained in the revised version of our manuscript.

b. Is SUPERFLEX faster than CLM? Some information on the running time for the two modelling approaches should be given.

Yes, it is much faster. Unfortunately, we did not save running time neither in our study nor in Rains's study and it appeared difficult to re-run the experiment. Although we do not have figures on that respect to add in the paper, an important point to mention however is that the experiment with superflex was carried out on a standard desktop pc within a few hours while an HPC (using 2 nodes of 12 Cores) was necessary to run Rains'experiment over a few days.

c. Why do we need a faster and less complex model? Which applications are addressed?
For climate applications, we don't need faster simulations, right?
I believe these questions should be addressed before the publication.

This has been highlighted and clarified in the new version of the article especially in the introduction:
Hence, a science question that is worth investigating is whether a flexible conceptual model, relying on parameter calibration, can reach the performance level of a more complex physically-based model for soil moisture simulations at large scales. […]
Faster models are key tools for carrying simulations at a large scale without implying a highomputational demand. Faster models are therefore powerful for near real-time forecasting applications and when large ensemble of model runs are required.

2) MAJOR: Is CLM calibrated on SMOS observations? As I believe it is not the case (from Rains et al., 2017), it is unexpected that CLM and SUPERFLEX perform similarly for the reproduction of SMOS brightness temperature (SUPERFLEX is calibrated on SMOS). Do the authors have an explanation? Similarly, results against in situ soil moisture observations are similar suggesting that CLM is performing good also without calibration. SUPERFLEX is calibrated with SMOS brightness temperature that in Australia is well correlated with in situ soil moisture (from previous studies), therefore I expect it works good against in situ soil moisture. On this basis, I believe CLM should be considered more reliable than SUPERFLEX. Can the authors comment on that?

SUPERFLEX is indeed calibrated using SMOS observations. Referee #1 is right; CLM is not calibrated using SMOS data. Indeed, while a conceptual model such as SUPEFLEX requires a calibration effort because its parameter values cannot be set a priori, CLM is not supposed to be calibrated as it is physically based and its parameters are derived from various input data describing the characteristics of the catchment. Moreover, one can argue that, because of a large number of parameters, calibrating CLM using SMOS data would not be an easy task especially due to the computational demand over a large basin such as the Murray Darling. The calibration of many parameters would lead to a widely reported equifinality issue.
We fully agree that CLM performs satisfyingly even without any calibration. However, we think it is important here to restate the objective of our study: we want to evaluate if a simplistic conceptual model when calibrated with a freely and globally available data product such as SMOS Tb can reach the performance level of a physically based model. We believe that the conclusions of our paper show that this is actually true. However, we do not agree with the statement that CLM should be considered more reliable than SUPERFLEX as the performance levels are rather similar for both models during the calibration and the validation period. The advantage of CLM is that it does not require any calibration, while the strength of SUPERFLEX is that it can reach the same level of performance when calibrated

with a freely and globally available SMOS data product. In addition, it can reach this performance level with a comparatively lower computational demand. We will clarify this as requested by Referee #1.

To clarify this, the following sentences are now written in the paper:

As a conclusion on the comparison between the forward run of both models, it can be highlighted that the two models finally reach similar performance levels when using as a reference either observed SMOS Tb or *in situ* measured soil moisture. It is also important to keep in mind that similar performance levels have been attained provided that the SFX model was calibrated whereas CLM was not calibrated using SMOS data. We argue that while a conceptual model such as SUPEFLEX requires a calibration effort because its parameter values cannot be set a priori, CLM is not supposed to be calibrated as it is physically based and its parameters are usually derived from various input data describing for example the characteristics of the catchment (Hou et al. 2012}. Moreover, one can argue that, because of a large number of parameters, calibrating CLM using SMOS data would not be an easy task especially due to the computational demand (Karagiannis et al.2019) over a large basin such as the Murray Darling. The calibration of many parameters would consequently lead to a widely reported equifinality issue. Of course it is worth mentioning here that CLM performs satisfactorily even without any calibration.

3) MODERATE: The differences between the open loop and the analysis are very small. Are they significant? Some tests to assess the significance of the obtained results should be performed.

To answer this question, we carried out and present Williams' significance tests in the revised version of the manuscript.

To assess the significance of the improvement in correlation as a result of the assimilation, we carried out Williams' significance tests (Williams, 1959). The null hypothesis in this test is that the improvement in correlation is not significant. For the upper soil layer the p-values are lower than 0.01 except for 2 stations where the correlation remains almost constant. For the deeper soil layer the p-values are lower than 0.01 except for 3 stations where the correlation remains almost constant or slightly decreased. This shows that the increase of correlation as a result of the assimilation of SMOS Tb can arguably be considered significant for most of the stations.

4) MODERATE: Does the SUPERFLEX model include lateral flow? If not (as I believe), it should be clarified.

This is right; In the simplified version used in this experiments SUPERFLEX does not explicitly simulate lateral flows within the root zone soil layers. This has been clarified in the revised version of the paper:
As a matter of fact, the distributed SFX implemented in this study does not simulate lateral flows within the root zone soil layers.

I listed in the specific comments a number of corrections and changes that are needed.
SPECIFIC COMMENTS (P: page, L: line or lines)
P1, Abstract: In the abstract, I have found too many details on the methodology and just few lines for the results. E.g., the results for simulating actual evapotranspiration are not mentioned. Please revise the abstract.

We thank Referee #1 for this relevant remark. We edited the abstract accordingly:

The main objective of this study is to investigate how brightness temperature observations from satellite microwave sensors may help in reducing errors and uncertainties in soil moisture and evapotranspiration simulations with a large-scale conceptual hydro-meteorological model. In addition, this study aims to investigate whether such a conceptual modelling framework, relying on parameter calibration, can reach the performance level of more complex physically-based models for soil

moisture simulations at a large scale. We use the ERA-Interim publicly available forcing dataset and couple the CMEM radiative transfer model with a hydro-meteorological model enabling therefore soil moisture, evapotranspiration and brightness temperature simulations over the Murray-Darling Basin in Australia. The hydro-meteorological model is configured using recent developments of the SUPERFLEX framework, which enables tailoring the model structure to the specific needs of the application as well as to data availability and computational requirements. The hydrological model is first calibrated using only a sample of SMOS brightness temperature observations (period 2010-2011). Next, SMOS brightness temperature observations are sequentially assimilated into the coupled SUPERFLEX-CMEM model (period 2010-2015). For this experiment, a Local Ensemble Transform Kalman Filter is used. Our empirical results show that the SUPERFLEX-CMEM modelling chain is capable of predicting soil moisture at a performance level similar to that obtained for the same study area and with a quasi-identical experimental set up using the CLM land surface model. This shows that a simple model, when calibrated using globally and freely available Earth observation data, can yield performance levels similar to those of a physically-based (uncalibrated) model. The correlation between simulated and \textit{in situ} observed soil moisture ranges from 0.62 to 0.72 for the surface and root zone soil moisture. The assimilation of SMOS brightness temperature observations into the SUPERFLEX-CMEM modelling chain improves the correlation between predicted and \textit{in situ} observed surface and root zone soil moisture by 0.03 on average, showing improvements similar to those obtained using the CLM land surface model. Moreover, the assimilation improves at the same time the correlation between predicted and \textit{in situ} observed monthly evapotranspiration by 0.02 on average.

P2, L5-10: In the introduction, the prediction of flood is mentioned but the modelling approach here developed is tested only in terms of soil moisture and evapotranspiration. Indeed, I expected to see also results in terms of river discharge simulations by reading the title (hydrological model). Anyhow, less emphasis should be given to flood forecasting in the introduction.

We thank Referee #1 for this relevant remark. We edited the introduction accordingly and focus it more on droughts.

P2, L32: Acronyms should be defined, and references to modelling approaches should be given. Throughout the text, the acronyms should be defined.

We defined the acronyms.

P3, L5: It should be "2009" instead of "2019".

Thanks for pinpointing us to this mistake

P3, L8: Please rename the "land surface modelling" for retrieval of soil moisture from SMOS brightness temperature. It makes confusion with land surface model. I would rename in "radiative transfer modelling".

This has been done

P4, L1: "tailoring the structure" of? Please clarify.

We edited the text so that it becomes clearer that the structure of the model (i.e. reservoirs…) can be tailored.

The SUPERFLEX modelling framework (Fenicia etal., 2016) enables tailoring the structure (i.e., adapt the model architecture via reorganising constituting reservoirs) for the specific needs of the application.

P5, Figure 1: Map of Australia should be smaller, and that of Murray-Darling bigger.

Figure 1 has been improved as requested by Referee #1

P5, L1: I understand the use of ERA-Interim for performing the simulation as in Rains et al. (2017); however, it would be highly interesting to test ERA5, the new ECMWF reanalysis.

We agree that this would be interesting in general. However, this goes beyond the scope of the paper as the new results would not allow for a meaningful comparison with the study by Rains et al (2017) which is one of the main objectives of this study. We are happy to consider this remark for further studies. We added the following sentence in the revised version of the manuscript:
It would have been possible as well to use the more recent and accurate ERA5 dataset, but we decided here to use ERA-Interim as it was also used in Rains et al. (2017).

P7, L5: It shouldn't be "surface" runoff, but total runoff, right?

This has been clarified: In the sentence, "surface runoff" was actually related to "routing function" (that simulates surface runoff):
The grey box in Figure2 also identifies the deeper reservoirs and the routing function that simulates subsurface and surface runoff based on deeper soil layer water storage.

P7, L9: not bold for "URI".

Thanks, this has been corrected.

P12, L14-15: Is the gradient of performance of SUPERFLEX similar to CLM? Please comment on that.
The gradient of performance of SUPERFLEX that is visible in Fig.3 is not observed in CLM's run. As argued in the paper:
A general gradient in the performance of SFX can be seen from the eastern to the western part of the basin whereas this gradient is not observed in the CLM performance. […] Considering that in our set up, the representation of the evapotranspiration is rather simplistic as it is based on the Hamon formula, this could explain the comparatively poor performance of the model in the western part of the basin.

P13, L1: Use "target" instead of "0" to avoid misunderstanding with the axis origin.
This has been modified in all Taylor diagrams

P14, L5-7: Why is there a strong underestimation of standard deviation? Do the authors have some explanations?

The following sentences have been added in the paper in that respect:
Our interpretation is that the two models are unable to reproduce the variance of SMOS observations mainly due to some limitations of the radiative transfer model. Indeed, even with completely dry or wet soils, the simulated Tb does not reach the extreme values of the SMOS Tb.

P16, L14: Why 25K2? Please add a reference or an explanation.

Here, we used the same value as in Rains et al. (2017) to keep the experiments comparable.

P16, L14: "average performance metric" as compared with? Please clarify.
The reference used for this comparison are the *in situ* soil moisture measurements. This is further clarified in the revised version of the paper:
Table 2 reports the spatially averaged performance metrics of the open loop (i.e., without assimilation) and the analysis SM simulations for two soil layer depths.

P19, L1-: This part should be moved to the method section.
This part has been moved accordingly in the revised version of the paper

P20, L13-14: Are we sure that ERA-Interim rainfall has larger errors for larger rainfall events?
This has been further explained in the paper:
Although there is no absolute evidence that errors are larger for larger rainfall events for the Murray Darling basin, this is something that was often reported (for different areas of interest) in the literature as for example in the study by Xu et al (2019).

P21, L9: This part should be moved to the method section.
This part has been moved accordingly in the revised version of the paper

P23, L6-17: There is no need to summarize the study in the conclusions; I suggest shortening this part.
The conclusion part has been shortened accordingly

**Referee #2**

The main objectives of this study is to compare a large-scale conceptual hydrometeorological model (SUPERFLEX) and a physically-based land surface models (CLM) in their ability to simulate SMOS-like brightness temperature (Tb) and soil moisture, and (ii) to evaluate the improvement in model predictions when assimilating SMOS Tb observations. It is well written and the abstract reflects the objectives and results well. Results are supported by appropriate figures and tables, references (some could be updated). This study is very interesting and promising. However, the paper does not do it justice. I feel like it was written quickly and that the authors skimmed over some key explanations. A more in-depth explanation of the methodology and analysis of the results are necessary. They are some caveats and I am particularly concerned about the title that does not reflect the real purpose of this study (comparison the ability of SUPERFLEX and CLM to simulate Tb and soil moisture).
We thank Referee #2 for this analysis. We put substantial efforts on better framing the objectives of the study and improving the explanations. The comparison between SFX and CLM is of course one of the objective but not the only one. However, we do not believe that including this information already in the title would be beneficial for the paper. We therefore adapted the title accordingly:
Assimilation of SMOS brightness temperature into a large-scale distributed conceptual hydrological model to improve soil moisture predictions: the Murray-Darling basin in Australia as a test case.
We better framed the objectives:
The main objective of this study is to investigate how brightness temperature observations from satellite microwave sensors may help in reducing errors and uncertainties in soil moisture and evapotranspiration simulations with a large-scale conceptual hydro-meteorological model. In addition, this study aims to investigate whether such a conceptual modelling framework, relying on parameter

calibration, can reach the performance level of more complex physically-based models for soil moisture simulations at a large scale. […]
The specific objectives of this study are as follows: (i) to compare the SUPERFLEX and CLM models in their ability to simulate Tb and soil moisture, and (ii) to evaluate the improvement in model predictions when assimilating SMOS Tb observations.
In general, we clarified the differences between our experiment and the one of Rains et al. (2017). We improved the presentation of the differences and similarities between the two experiments.

To me this kind of comparison is a bit unfair from the beginning as you do not necessarily want a large-scale distributed conceptual hydrological model and a large scale physically based land surface model (CLM) for the same purpose (?)

We agree with Referee #2 on the fact that one do not necessarily want a large-scale distributed conceptual hydrological model and a large scale physically based land surface model (CLM) for the same purpose. We clarified this:
Faster models are key tools for carrying simulations at a large scale without implying a highomputational demand. Faster models are therefore powerful for near real-time forecasting applications and when large ensemble of model runs are required.

Not to mention the fact that SUPERFLEX is calibrated, what about CLM? If authors wish to pursue in this way, then I am missing a proper description of the CLM set up (not only referring to a previous study and mentioning a 'quasi-identical' set up). My recommendation is major review, please find below an attempt to help.
We also clarified the fact that SFX is calibrated and CLM not and we explained why and we further discussed the implications of this. More information on CLM is provided in the updated version of the paper. SUPERFLEX is calibrated whereas CLM is not. Indeed, while a conceptual model such as SUPEFLEX requires a substantial calibration effort because its parameter values cannot be set a priori, CLM is not supposed to be calibrated as it is physically based and its parameters can be determined based on the basin's known characteristics. Moreover, one can argue that calibrating CLM using SMOS data would not be an easy task, especially because of a high number of parameters and a significant computational effort needed to achieve a robust calibration for a large basin such as the Murray Darling. We clarified this throughout the article:
From the SFX model description, it appears that this conceptual model is much simpler than CLM in particular for the following reasons:
- Whereas CLM estimates latent heat fluxes between soil and atmosphere based on aerodynamic diffusion equation and Monin-Obukhov similarity theory (Oleson et al., 2013), taking into account many information such as soil and vegetation type, surface roughness, atmospheric stability, the vegetation coverage, SUPERFLEX lumps energy balance contribution to water balance via a simpler potential evapotranspiration formula, namely the Hamon formula.
- In this study, SFX is structured with two soil layers (respectively 0-7cm and 7-21cm) whereas CLM considers five layers for the same soil depth range.
- The set up of CLM therefore requires much more input data (e.g., soil types and land use), and Superflex has a limited number of parameters (6 parameters per cell) compared to CLM (potentially tens of parameters per cell, Hou et al, 2012}.
It is worth noting here that this list is not exhaustive and that many other processes are further simplified in SFX compared to CLM. A detailed presentation of CLM is available in Oleson et al. (2013). […]
As a conclusion on the comparison between the forward run of both models, it can be highlighted that the two models finally reach similar performance levels when using as a reference either observed SMOS Tb or *in situ* measured soil moisture. It is also important to keep in mind that similar performance levels have been attained provided that the SFX model was calibrated whereas CLM was not calibrated

using SMOS data. We argue that while a conceptual model such as SUPEFLEX requires a calibration effort because its parameter values cannot be set a priori, CLM is not supposed to be calibrated as it is physically based and its parameters are usually derived from various input data describing for example the characteristics of the catchment (Hou et al. 2012}. Moreover, one can argue that, because of a large number of parameters, calibrating CLM using SMOS data would not be an easy task especially due to the computational demand (Karagiannis et al.2019) over a large basin such as the Murray Darling. The calibration of many parameters would consequently lead to a widely reported equifinality issue. Of course it is worth mentioning here that CLM performs satisfactorily even without any calibration.

General major comments (additionally to what is mentioned above)
-From the abstract I see scores and headlines but I have no clue where the study takes place, please introduce south eastern Australia from the beginning (maybe from the title, it has to change anyway to reflect the content of the work).

We thank Referee #2 for this relevant remark. We edited the abstract and the title accordingly
Title:
Assimilation of SMOS brightness temperature into a large-scale distributed conceptual hydrological model to improve soil moisture predictions: the Murray-Darling basin in Australia as a test case.
Abstract:
We use the ERA-Interim publicly available forcing dataset and couple the CMEM radiative transfer model with a hydro-meteorological model enabling therefore soil moisture, evapotranspiration and brightness temperature simulations over the Murray-Darling Basin in Australia.

- Also from the abstract it is surprising that ERA-Interim is still used rather than ERA5. The recent literature (2018, 2019) is already full of studies demonstrating the added value of ERA5 with respect to ERA-Interim. I assume that in the previous CLM study (Rains et al., 2017) ERA-Interim was used and that is the rational for keeping it. This should be stated somewhere and ERA5 mentioned, if not tested as I believe it will prove useful.

We agree that ERA 5 is a better dataset than ERA interim and is of great interest. However, as acknowledged by Referee #2, using ERA5 for this study would render any comparison with the study by Rains et al. (2017) rather meaningless. We consider this comparison to be one of the main objectives of this study. We will better explain why we used ERA-interim and introduce Era5 in the paper. We added the following sentence in the revised version of the manuscript:
It would have been possible as well to use the more recent and accurate ERA5 dataset, but we decided here to use ERA-Interim as it was also used in Rains et al. (2017).

- work must be done on statistical scores to provide an indication of how significant they are, I suggest to add at least p-values to assess the significance of each datasets and a 95% confidence interval (based on boot strapping?) to assess either or not differences from the 2 configurations are significant (that can possibly hamper your conclusions?).

To answer this question, we carried out and present Williams' significance tests in the revised version of the manuscript.
To assess the significance of the improvement in correlation as a result of the assimilation, we carried out Williams' significance tests (Williams, 1959). The null hypothesis in this test is that the improvement in correlation is not significant. For the upper soil layer the p-values are lower than 0.01 except for 2 stations where the correlation remains almost constant. For the deeper soil layer the p-values are lower than 0.01 except for 3 stations where the correlation remains almost constant or slightly decreased. This shows that the increase of correlation as a result of the assimilation of SMOS Tb can arguably be considered significant for the large majority of stations.

-Figures and tables should be self explanatory (?) please expand captions, add units when necessary (Kelvin: : :), label each panels for sake of clarity and refer to the labelling in the captions (some figure are hardly visible).

We clarified and improved the figures and captions accordingly.

Other comments - scores from the abstract should more detailed, are you talking about surface soil moisture? Root zone soil moisture?

The scores given in the abstract have been computed based on observed soil moisture for both the surface layer and the 30cm root zone soil layer. This has been clarified in the revised version of the article:
The correlation between simulated and *in situ* observed soil moisture ranges from 0.62 to 0.72 for the surface and root zone soil moisture. The assimilation of SMOS brightness temperature observation into the SUPERFLEX-CMEM modelling chain improves the correlation between predicted and *in situ* observed surface and root zone soil moisture by 0.03 on average showing improvements similar to those obtained using the CLM land surface model

- P.2, L.13, '[...] as uncertain forcing [...]' OK so justify the use of ERA-Interim over ERA5

We agree that ERA 5 is a better dataset than ERA interim and is of great interest. However, as acknowledged by Referee #2, using ERA5 for this study would render any comparison with the study by Rains et al. (2017) rather meaningless. We better explained why we used Era-interim and introduce Era5 in the paper:
It would have been possible as well to use the more recent and accurate ERA5 dataset, but we decided here to use ERA-Interim as it was also used in Rains et al. (2017).

- P.2, L.27, surface soil moisture (SSM)

We added « surface » before soil moisture accordingly.

- P.3, L.5, November 2019 ? Do you mean 2009 ?

This is right . Thanks for highlighting this mistake.

- P.14, L.15, "[: : :] is impacting soil moisture variations more significantly [...]" what is the meaning of "significantly"?

Here we meant to say that evapotranspiration has a higher impact on soil moisture variations. This has been clarified:
[…] where evapotranspiration has a higher impact on soil moisture variations.

- Figure 2 must be improved, ground based measurement stations are barely visible, also is the main river represented the only one in Australia (this is not a paper quality figure).

Our assumption is that Referee #2 is referring to figure 1 rather than figure 2. The right panel of this figure has been enlarged to render the measurement stations more visible. We improved this figure in general.

- section 2.2.1, if this work has been published elsewhere, maybe it can be put in an annexe / supplementary ?

To our knowledge this work has not been published elsewhere

- P.7, L.2, "[: : :] 0.25_ matching the one used in ERA-Interim dataset." misleading at you put is at 0.25_ while its native spatial resolution is closer to 80km

We thank Referee #2 for highlighting this. We clarified this point in the revised version of the paper: The model is therefore distributed over grid cells of 0.25° aligned on the grid used in the ERA-Interim dataset and simulations are carried out at an hourly time step.

- I am missing somewhere a clear description of the 2 models set up

Section 2.2.1 (The conceptual hydrological Model) is dedicated to the SFX set up description. A description of the main differences between CLM and SFX has been added in the updated version of the paper:
From the SFX model description, it appears that this conceptual model is much simpler than CLM in particular for the following reasons:
- Whereas CLM estimates latent heat fluxes between soil and atmosphere based on Monin-Obukhov similarity theory (Oleson et al. 2013), taking into account many information such as the types of soil and vegetation and the vegetation status, the vegetation coverage, SUPERFLEX lumps energy balance contribution to water balance via a simpler potential evapotranspiration formula, namely the Hamon formula.
- In this study, SFX is structured with two soil layers (respectively 0-7cm and 7-21cm) whereas CLM considers five layers for the same soil depth range.
- The set up of CLM therefore requires much more input data (e.g., soil types and land use), and Superflex has a limited number of parameters compared to CLM.
It is worth noting here that this list is not exhaustive and that many other processes are further simplified in SFX compared to CLM. A detailed presentation of CLM is available in Oleson et al. (2013).

- Section 2.2.3, please discuss further the possible impact on the 2 models comparison.

We put efforts on clarifying in the paper that SUPERFLEX is calibrated and CLM not. We also emphasised the fact that the calibration of Superflex using SMOS data gives it a higher chance to reach better performance levels. Moreover, we argue that calibrating CLM using SMOS data would not be an easy task, especially due to the required computational demand for calibrating a very high number of parameters over a large basin such as the Murray Darling:
As a conclusion on the comparison between the forward run of both models, it can be highlighted that the two models finally reach similar performance levels when using as a reference either observed SMOS Tb or *in situ* measured soil moisture. It is also important to keep in mind that similar performance levels have been attained provided that the SFX model was calibrated whereas CLM was not calibrated using SMOS data. We argue that while a conceptual model such as SUPEFLEX requires a calibration effort because its parameter values cannot be set a priori, CLM is not supposed to be calibrated as it is physically based and its parameters are usually derived from various input data describing for example the characteristics of the catchment (Hou et al. 2012}. Moreover, one can argue that, because of a large number of parameters, calibrating CLM using SMOS data would not be an easy task especially due to the computational demand (Karagiannis et al.2019) over a large basin such as the Murray Darling. The calibration of many parameters would consequently lead to a widely reported equifinality

issue. Of course it is worth mentioning here that CLM performs satisfactorily even without any calibration.
- Figure 3, units, significance, labels

This has been done

- Table 1, as it stands it is not very useful, expand the caption so readers may know what is it about, statistics between what and what? What are the units? Cal stands for calibration, Val stands for validation...(general major comment), use same number of digit..

We improved the caption of this table and the table itself:
Performance and error metrics of Superflex simulated Tb using as a reference SMOS Tb: Pearson's correlation coefficients (left hand side panels), unbiased root mean square deviations (ubRMSD, center panels) and Mean Biases (right hand side panels) between SFX-CMEM predictions and SMOS observations of Tb during the calibration (top panels) and the validation (bottom panels) periods.

- Figure 4, is left panel useful? Significance of the differences in figure 5?

Yes as the left panel proposes average values that are more integrative, we believe it is useful. For the significance, we also computed p-values of the correlations:
The p-values associated to Pearson's correlation coefficients are all below 0.01 lending therefore weight to the significance of the correlation between simulated Tb and SMOS Tb.

- P.15, L.6, "[: : :] time series [...]" a figure would prove useful

A figure with a time series would only show results for one station. We believe that showing a more general analysis using Taylor diagrams is more relevant.

- Figure 6, same min/max for axis of left and right panels

Since the considered soil depths differ, the performance value ranges in different intervals. We do not see the benefit of "same min/max for axis of left and right panels" as this would render the figure less readable (i.e. the points will be closer to each other).

- P.17, L.9-12, please discuss the use of SMOS anomalies as it could be an explanation (?)

The fact that we assimilate SMOS anomalies helps reducing bias between simulations and observations prior to the assimilation. However, we do not see why the assimilation of anomalies would explain the fact that ubRMSE and RMSE (computed based on in situ soil moisture observation) are not reduced. Moreover, many recent studies actually assimilate anomalies in order reduce bias between model and satellite observation. The following paragraph of the paper explain this choice:
As mentioned in many studies dealing with the assimilation of satellite SM or Tb (e.g. Al Bitar et al., 2012; Matgen et al., 2012; Rains et al., 2017; Al-Yaari et al., 2017), bias removal prior to the assimilation is often a necessary step. In our study, we reduce the bias between simulations and observations by deriving model and observation anomalies, following an identical approach to the one described in Rains et al. (2017).

- Figure 8, I do not understand bottom left panel, rainfall and #obs? Also why the number of stations differs from a panel to another?

In the bottom left panel, the same colour scale is used to indicate rainfall amounts (map) and number of SMOS observations assimilated at each measurement station (colour of each points). This has been clarified in the new version of the paper. The number of stations differs from one panel to another as all stations do not measure soil moisture for all soil depths. The bottom panel indicates all measurement stations.

The top panels show stations measuring SM in the 8 and 30 cm top soil layer. This has been clarified as well.

To analyse the spatial distribution of correlation improvement as a result of the SMOS data assimilation, Figure 8 maps the changes in correlation between predictions and *in situ* measurements of soil moisture at all stations and the temporal average of assimilation SM increments (top panels), together with the average annual rainfall and Ep and the number of SMOS records assimilated over *in situ* soil moisture measurement sites (bottom panels). The two top panels show local correlation improvements and SM increments in the 8 (top left panel) and 30 cm (top right panel) top soil layer respectively. The bottom panels in Figure 8 show the climate variability over the Murrumbidgee catchment using as a proxy the average annual rainfall and Ep (data provided by the Australian Bureau of Meteorology), together with the number of SMOS records assimilated over in situ soil moisture measurement sites. In the bottom left panel of Figure 8, the same colour scale is used to indicate the rainfall amount (map) and the number of SMOS observations assimilated at each measurement station (colour of each points). The number of stations differs between left and right top panels as all stations do not measure soil moisture for all soil depths. The bottom panel indicates all measurement stations.

- Figure 9 is interesting!

We thank Referee #2 for this positive remark.

- Figure 10, not clear enough that 5 pairs of data are represented on the Taylor Diagram, pleas improve the quality.

We improved this figure

- Title presents 1 objective, the abstract 2 and the beginning of the conclusion 3, please be consistent.

We modified the title accordingly:
Assimilation of SMOS brightness temperature into a large-scale distributed conceptual hydrological model to improve soil moisture predictions: the Murray-Darling basin in Australia as a test case.
We modified the abstract accordingly:
The main objective of this study is to investigate how brightness temperature observations from satellite microwave sensors may help in reducing errors and uncertainties in soil moisture and evapotranspiration simulations with a large-scale conceptual hydro-meteorological model. In addition, this study aims to investigate whether such a conceptual modelling framework, relying on parameter calibration, can reach the performance level of more complex physically-based models for soil moisture simulations at a large scale. […]
We modified the conclusion accordingly:
This study introduced and evaluated a large-scale SM modelling chain that is based on and takes advantage of the assimilation of SMOS Tb into a spatially distributed conceptual hydrological model coupled with a radiative model transfer. We assessed the performance of such a modelling chain and its associated data assimilation system and compared it with that of a quasi-identical set up using the physically based CLM land surface model (Rains et al., 2017). We evaluated therefore whether a SM modelling chain, based on a conceptual hydrological model, is able to reach the same performance level as that of one based on a physically-based model, the main advantage of a conceptual model being its substantially lower computational demand. Eventually, we also evaluated how the assimilation of SMOS Tb can help in improving evapotranspiration predictions.[…]

- I am personally not a big fan of bullet points in a conclusion but I may be a personal statement.

We thank Referee #2 for this remark.

**Referee #3**

I fully agree with the author to try out the usability of a conceptual hydrological model (when compared to detailed LSMs) assimilating TB, in terms of capturing hydrological processes at large scale. Such concept of spatially calibration (or DA to enable optimal estimate) is very innovative to overcome certain calibration problems of distributed hydrological model with only one stream flow gauge.

We thank Referee #3 for this assessment.

I do have several comments for the authors to consider:
1. The manuscript focused on SFX model simulations and their coupling with CMEM as well as the whole setup in the DA framework, as well as the results comparison with the previous study using CLM. Although it is understandable that the author try to maintain the quasi similar setup to enable the consistent comparison, the current presentation of such 'similarity' is not clear. Lots of details need to dig out by readers via reading the paper of Rains. This reviewer is wondering if the author can make better presentations on this perspective, either via a diagram, via summary tables, etc.

We thank Referee #3 for highlighting this. We will put efforts on further explaining and clarifying the similarities and the differences between the two experiments. A description of the main differences between CLM and SFX has been added in the updated version of the paper:
From the SFX model description, it appears that this conceptual model is much simpler than CLM in particular for the following reasons:
  • Whereas CLM estimates latent heat fluxes between soil and atmosphere based on Monin-Obukhov similarity theory (Oleson et al. 2013), taking into account many information such as the types of soil and vegetation and the vegetation status, the vegetation coverage, SUPERFLEX lumps energy balance contribution to water balance via a simpler potential evapotranspiration formula, namely the Hamon formula.
  • In this study, SFX is structured with two soil layers (respectively 0-7cm and 7-21cm) whereas CLM considers five layers for the same soil depth range.
  • The set up of CLM therefore requires much more input data (e.g., soil types and land use), and Superflex has a limited number of parameters compared to CLM.
It is worth noting here that this list is not exhaustive and that many other processes are further simplified in SFX compared to CLM. A detailed presentation of CLM is available in Oleson et al. (2013).

2. It is very good to see the forward simulation comparisons between SFX-based and CLM-based models. On the other hand, the advantage of assimilating SMOS TB on soil moisture and ET is not presented in a satisfied manner. The manuscript only shows the performance for the in-situ sites, but not the spatial pattern changes of soil moisture and ET (before and after assimilating Tb). This point should be addressed to show straightforwardly the advantage of incorporating spatial information for distributed conceptual hydrological model simulations.

According to Referee #3's suggestion, we added in figures 8 and 9 assimilation increments in the revised version of the paper to investigate spatially the effect of Tb assimilation SM and ET.

3. It is not clear if the author only run the model at the timestep of satellite observation intervals or also run the model in between observations? E.g., if the model only start to run with the satellite

observation when they are available, then the author missed a lots of details in between satellite observations It is assumed with the forcing the model can get more frequent simulation results as outputs to capture hydrological processes?

We apologize that this was not clear enough. The ensemble of models run at an hourly time step as well in-between SMOS observations. We clarifed this point in the revised version of the article:
The model is therefore distributed over grid cells of 0.25\degree aligned on the grid used in the ERA-Interim dataset and simulations are carried out at an hourly time step.

4. Last but not least, it is strongly recommended to have a native English editor to go through the manuscript. Some specific comments please find attached PDF.

According to Referee #3's suggestion, we carefully proofread the paper before its resubmission.

The Specific comments in the supplementary review material provided by Referee 3 were as well carefully addressed in the new version of the paper.

Specific remarks from the pdf:
So, there is no ensemble of model simulations between two satellite observations?

Of course there is. As written in the paper :
Each time a SMOS observation is available, the SUPERFLEX state variables related to the water content in the various soil layers are updated and the model simulations are resumed until the next SMOS observation becomes available.
This means that the simulations are carried out in-between two assimilation events

It depends on if you have more frequent simulations or just use the satellite observation intervals, right?

Simulations are carried at an hourly time step. So the model results are more frequent than SMOS observation

The assimilation of SM over high latitude cold region is also important to be mentioned, since it is hard to have in-situ SM observation over there.

Su, Z., de Rosnay, P., Wen, J., Wang, L. and Zeng, Y. (2013) Evaluation of ECMWF's soil moisture analyses using observations on the Tibetan Plateau : open access. Journal of geophysical research : D: Atmospheres, 118:. 5304-5318.

We added the suggested reference in the paper

this should not be here. Different models will not lead to different model forcings ....

Forcing is forcing!

If the models are different, they potentially use different forcing datasets. We clarified this in the revised version of the paper:

However, the land surface model used for the SM retrieval and the model used for the background simulation are often different for example in terms of process representation, model structure and model forcing datasets (e.g., air and soil temperature)

very confusing sentences here. Please rephrase.

This has been done
This potentially result in inconsistencies in the way SM is simulated by the model and retrieved from the observation. Moreover, De Lannoy et al. (2016) argued that this issue can lead to correlation between retrieved and simulated SM errors that cannot be easily handled by data assimilation filters.

Tb is Tb, 'like' is not a scientific word.

We argue that Tb depends on the sensor characteristics (optical, microwave… different frequencies…..). Therefore Tb is not necessarily unique as stated by Referee #3. As a consequence, we prefer making sur that it is clear from the paper. However as Reviewer #3 does not like the word "like", we removed it from the paper. We added the following sentence to clarify this:
It is worth mentioning that, here and in the remainder of the paper "`simulated Tb"' is used for naming Tb that is derived, using the radiative transfer model, from the simulated soil moisture. The simulated Tb is therefore meant to emulate SMOS observation based on simulated soil moisture.

Do you mean you just neglect this part of processes? Or just not describing them here? they do affect the soil moisture as the bottom boundary conditions!

The deeper reservoirs are actually switched off and therefore not described in the paper. In Superflex, as explained in the paper, water only flows from Sur2 to deeper reservoirs and not backward.

In this study, since we focus on the two upper root zone layers that are of interest for simulating soil moisture, the deeper reservoirs and the routing function are switched off and not further referred to in the remainder.

This is ok, but then the comparison of spatial soil moisture pattern before and after assimilating the Tb observation (for both depths) would be more convincing.  As such, we will see if the part not well simulated by SFX is improved or deteriorated? why the spatial soil moisture pattern was not shown?

The map of the temporal average of the soil moisture absolute increment of is now shown on figure 8

the spatial pattern of ET should be shown with the difference before and after assimilating Tb.

The map of the temporal average of the evapotranspiration absolute increment is now shown on figure 9

[revised manuscript text omitted]

---

## Author Response (AR2)

This document contains our answers to the referees' and editor's comments. We would like first to thank Referees #1, 2 and 3 and the editor for the careful reading of the paper and their relevant remarks and comments. In the remainder, the Referees' remarks are written in black while our answers are written in blue. Moreover, cited text from the revised version of the paper is written in red.

Enclosed is as well our revised paper

**Editor:**

Your manuscript "Assimilation of SMOS brightness temperature into a large-scale distributed conceptual hydrological model to improve soil moisture predictions: the Murray-Darling basin in Australia as a test case" has been subjected again to review by the original three reviewers. Two of them recommended now acceptance of the manuscript and one reviewer recommends major revision.

The paper still lacks details on the complexity of the two models and the differences between the models. Also some further important details are needed like the handling of the temporal model input. The small improvements related to data assimilation need to be discussed more critically. I suggest moderate revision of the paper.

In your answer to the main points and detailed comments, please indicate how comments have been handled exactly, indicating also whether text has been deleted and what the position of newly included text blocks is. I am looking forward to the new version of the paper.

According to the editor's feedback, we provided further details on the complexity of the two models, on the differences between the models and on the handling of the temporal model input. We also discussed more critically the moderate improvements related to data assimilation (please see answers to Referee #3's comments).

**Referee #1:**

The authors successfully addressed all reviewers' comments. Specifically, the authors clarified the set up of the two modelling approaches and their differences. I have two remaining minor comments: 1) I would add some details on the computational demand of the two modelling approaches. 2) If possible, I would include the code of SFX model that can be very useful for the hydrology readers interested to implement the model.

We thank Referee #1 fort his positive feedback.

We propose to provide the Superflex model code on request, as indicated now in the paper (page 26, lines 9-10):

"Code availability. The code of the version of the Superflex model used for this study can be obtained upon request to the corresponding author."

Moreover, we add some details on the computational demand of the two modelling approaches, as requested (page 26, lines 3-5):

"While our assimilation experiment with SFX was carried out on a Personal Computer within a few hours, a High-Performance Computing cluster (using 2 nodes of 12 Cores) was necessary to run the

Rains et al. (2017) experiment over a few days. This shows the added value of a computationally efficient conceptual model, especially for applications where computational time is critical."

**Referee #2:**

Accepted as is

We thank Referee #2 for this positive feedback.

**Referee #3**

1. The author only provided some brief points to explain CLM is more complex than SFX model. On the other hand, it is not specifically clear what are difference in complexity, in terms of physical processes. This matters, particularly, for soil moisture predictions. This reviewer does understand it is not needed to present all the difference between two models, while still believe a tabulated comparison is needed, especially for those parts matter the most to the discussions of results (ET, SM, etc.). This is to verify the claim in the abstract "Our empirical results show that the SUPERFLEX-CMEM modelling chain is capable of predicting soil moisture at a performance level similar to that obtained for the same study area and with a quasi-identical experimental set up using the CLM land surface model". This reviewer cannot understand how 'quasi-identical' it is.

We thank Referee #3 for this feedback, remarks and suggestions to further improve the manuscript.

Two tables have been added in the revised version of the manuscript to further clarify the differences between processes implemented, respectively, in CLM and Superflex and the minor differences between the two experiments. Table 1 (page 9) summarizes the main processes implemented in CLM and SFX. It allows therefore to assess the markedly different level of complexity of the two models. Table 2 (page 12) further describes how quasi identical the two experiments are. It clearly identifies the minor differences between the two assimilation experiments.

2. 'In this study, since we focus on the two upper root zone layers that are of interest for simulating soil moisture, the deeper reservoirs and the routing function are switched off and not further referred to in the remainder.' The above treatment means a twist of the physical processes of SFX. This reviewer is interested how this treatment will affect the final assimilation results.

The removal of the deeper reservoirs of SFX has no effect on the soil moisture simulations as in Superflex there is no upward water circulation from the deeper reservoirs to the upper ones. As a matter of fact, when deeper reservoirs are switched off, water exits root zone soil layers based on the usual equations. The soil moisture simulations within both root zone reservoirs are therefore not impacted. This information has been added in the revised version of the manuscript (page 7, lines 14-17):

"It is worth mentioning that the removal of the deeper reservoirs of SFX has no effect on the soil moisture simulations as in SFX there is no upward water circulation from the deeper reservoirs to the upper ones. As a matter of fact, when deeper reservoirs are switched off, water exits root zone soil layers based on the usual equations. The soil moisture simulations within both root zone reservoirs are therefore not impacted."

3. "The model is therefore distributed over grid cells of 0.25 aligned on the grid used in the ERA-Interim dataset and simulations are carried out at an hourly time step." This reviewer is very curious on how

this is possible. ERA-Interim only provides four analyses per day, at 00, 06, 12 and 18 UTC, while the simulation is carried out at an hourly time step. Is there certain interpolation involved and how?

For the ERA-Interim accumulated variable (i.e., rainfall) the predicted amount is redistributed uniformly from 6h accumulation to 1h accumulation in order to keep water balance. For the other variables (i.e., air and soil temperature), the value is imposed constant over 6h and equal to the era-interim predicted value. This clarification has been added to the manuscript (page 5, lines 12-14):

"For the accumulated variable (i.e., rainfall) the predicted amount was redistributed uniformly from 6h accumulation to 1h accumulation in order to keep water balance. For the other variables (i.e., air and soil temperature), the value was imposed constant over 6h and equal to the era-interim predicted value."

4. For me the improvement of correlation coefficient with 0.03 (for SM) and 0.02 (for ET) do not seem significant, even though William's significance test was carried out. Does it mean that the assimilation of TB does not matter that much? Or please help to discuss with similar studies, or compare with those studies using spatial information (soil moisture spatial info) to do calibration while not using DA.

We agree with Referee #3 that the values of correlation improvement are moderate for soil moisture and rather low for Et, as acknowledged in the paper. However, they are still statistically significant, as testified by the William's significance test. This test was indeed proposed as a tool for evaluating if an increase of correlation is statistically significant or not.
For the soil moisture correlation improvement, the first and maybe main point to consider in our opinion is the already high correlation values (0.77 and 0.7 for the two layers) of the open loop (before the assimilation) that leaves less room for improvement compared to other studies such as Rains et al (2017) or De Lannoy and Reichle (2016a) that reported substantially lower background correlation (around 0.6). Moreover, the fact that the SFX model is already calibrated using SMOS Tb and ERA interim data can also explain the fact that the improvement of the soil moisture predictions are slightly lower, but of the same order of magnitude, than in the other studies relying on uncalibrated land surface models. We would further argue that the limited effect of the Tb assimilation is in itself an interesting result for the community that we try to address with our study.
This is now further developed in the manuscript (page 19, lines 6-15):
"As can be seen in Table 4, the assimilation allows for a moderate increase in rho for the two soil layers depicted in the model when comparing observed and simulated soil moisture time series. Specifically, the rho increases on average by more than 0.03 for both soil layer depths. These improvements are similar, although slightly lower to those obtained in the study by Rains et al. (2017, experiments DA2 and DA0), namely ca. 0.06 for upper layers and 0.03 for deeper layers. One possible explanation for the slightly lower improvements in rho for the top-layer can be found in the SFX open-loop performance being already higher (rho=0.77) than that of CLM (rho=0.61). This arguably limits the room for improvement as a result of the assimilation as the SFX-based open-loop outperforms the one based on CLM. Moreover, the fact that the SFX model was calibrated using SMOS Tb and forced using the ERA-interim dataset can also explain the fact that the improvement of the soil moisture predictions are slightly lower than in other studies relying on uncalibrated land surface models (e.g., Rains et al., 2017; De Lannoy and Reichle, 2016a)."
For the ET improvement, it is clearly acknowledged that the correlation improvements are limited except for one station (page 23, line 6-8):
"The effect of the assimilation on evapotranspiration is substantially positive for one station, limited for 3 of them and slightly negative for the last one. As for SM, one can notice in the right panel of Figure 10 that the rho is further improved when absolute evapotranspiration increments tends to be higher."

5. 'highomputational demand' ◊ 'high computational demand'
This typo was only visible in the version of the manuscript with track change. We made sure that this is correctly written now.

6. In Figure 8 "maps of the number of assimilated SMOS Tb observation at each soil moisture measurement station, in relation with inter-annual average rainfall (bottom left panel) and inter-annual average Ep (bottom right panel)." This reviewer cannot see the number of assimilated SMOS Tb in this figure, and a bit confused by the caption here.

The same colorscale is used for showing two different pieces of information at the same time, namely the number of satellite observations assimilated into the model together with the inter-annual average rainfall. While the number satellite observations assimilated into the model is indicated by the colours of the dots, the rainfall and Ep are indicated via the continuous colour map. We clarified this in figure 8 caption of the revised version of the manuscript:

[revised manuscript text omitted]

---

## Author Response (AR3)

This document contains our answers to the referees' and editor's comments. We would like first to thank the referee and the editor for the careful reading of the paper and their relevant remarks and comments. In the remainder, the Referee' remarks are written in black while our answers are written in blue. Moreover, cited text from the revised version of the paper is written in red.

Enclosed is the revised version of our paper.

**Editor:**

The new version of your paper was reviewed by one remaining reviewer. I am happy to communicate you that only some technical corrections have to be made for the publication of the manuscript.

The technical corrections suggested by the reviewer have been done.

**Reviewer:**

Thanks a lot for the authors' efforts to improve the clarity of this manuscript.
We thank the reviewer for this positive feedback.

I only had the following comments for further consideration:

1. "…As a matter of fact, when deeper reservoirs are switched off, water exits root zone soil layers based on the usual equations. ..." what do you mean 'usual equations'. Please be specific.

We provided further details in the paper on that respect:

"As a matter of fact, when deeper reservoirs are switched off, water exits root zone soil layers based on the usual percolation and/or subsurface flow equations (e.g., Eq. 1)."

2. Table 1 For 'Surface Energy Fluxes', in CLM, the author stated that 'Air and soil heat fluxes (Monin-Obukhov similarity)', which seems a bit confusing.

We thank the reviewer for this relevant suggestion. We improved and clarified table 1 as well as the associated terminology.

- It is understandable that MO theory was used to calculated the MO length L, which is further applied to calculate aerodynamic resistances. However, if you say surface energy fluxes, it is already referred to ground surface sensible heat flux, laten heat flux, ground heat fluxes etc. There are differences in terms of vegetated surface or bare soil and so on though. If feasible, please list the most important equations to avoid ambiguity. The Table 1 as it stands now seems not informative and not precise.

To clarify this, we separated radiative fluxes and energy fluxes in Table 1 (see below). It is clearer now that evaporation and transpiration are estimated via energy fluxes. Moreover, we separated the ground and vegetation latent heat fluxes computation in CLM for the sake of clarity, as suggested by the reviewer.

- 'Soil Heat Fluxes' are not exactly correct. Perhaps Soil Surface or Ground Surface, but then, you already mentioned surface energy fluxes

To correct this, we removed the term "soil heat flux" and only mention energy fluxes in Table 1 (see below).

- 'air' I understand the author would like to imply that there is a land-atmosphere interaction. However, 'air flux' is really not what CLM will do. Usually 'air flux' can be provided by atmospheric models, and still they don't call it 'air flux' as such ... ...

We corrected this (see table 1 below) and thank the reviewer for highlighting it.

**Table 1.** Main processes implemented in the CLM- and SFX-based studies.

| Main Processes | CLM | | SFX |
|---|---|---|---|
| Radiative fluxes | Solar and long wave radiations | | Lumped evapotranspiration (Hamon formula) |
| Evaporation & transpiration | Ground latent heat flux / Vegetation latent heat flux | (Monin-Obukhov similarity) | |
| Subsurface vertical flow | Across 10 soil layers (adapted Richards equation) | | Across 2 soil layers (power low dynamics) |
| Subsurface lateral Flow | Only in the saturated groundwater | | - |